



# Assessment of JSBACHv4.30 as land component of ICON-ESM-V1 in comparison to its predecessor JSBACHv3.2 of MPI-ESM1.2

Rainer Schneck[1], Veronika Gayler[1], Julia E.M.S. Nabel[1,2], Thomas Raddatz[1], Christian H. Reick[1], and Reiner Schnur[1]

[1]Max Planck Institute for Meteorology, Hamburg, Germany
[2]Max Planck Institute for Biogeochemistry, Jena, Germany

**Correspondence:** Rainer Schneck (rainer.schneck@mpimet.mpg.de)

**Abstract.** We assess the land surface model JSBACHv4, which was recently developed at the Max Planck Institute for Meteorology as part of the effort to build the new Earth System model ICON-ESM. We assess JSBACHv4 in simulations coupled with ICON-A, the atmosphere model of ICON-ESM, hosting JSBACHv4 as land component to provide the surface boundary conditions. The assessment is based on a comparison of simulated albedo, Land Surface Temperature (LST), Leaf Area
Index (LAI), Terrestrial Water Storage (TWS), Fraction of Absorbed Photosynthetic Active Radiation (FAPAR), Net Primary Production (NPP), and Water-Use-Efficiency (WUE) with corresponding observational data. JSBACHv4 is the successor of JSBACHv3, therefore, another purpose of this study is to document how this step in model development has changed model biases. This is achieved by also assessing in parallel results of coupled land-atmosphere simulations with the preceding model ECHAM6 hosting JSBACHv3.

Large albedo biases appear in both models over ice sheets and in central Asia. The temperate to boreal warm bias observed in simulations with JSBACHv3 largely remained in JSBACHv4, despite the very good agreement with observed LST in the global mean. For the assessment of changes in land water storage, a novel procedure is suggested to compare the gravitational data from the GRACE satellites to simulated TWS. It turns out that the agreement of changes in the seasonal cycle of TWS is sensitive to the representation of precipitation in the atmosphere model. The LAI is generally too high which is partly caused
by too high soil moisture but also by the parameterization of the phenology itself. The pattern of WUE is for both models largely as observed. In India WUE is too high probably because JSBACH does not incorporate irrigation in our simulations. WUE differences between the two models can be traced back to differences in precipitation patterns in the two coupled land-atmosphere simulations. For both models, most NPP biases can be associated with biases in water stress, LAI and FAPAR. Particularly the NPP bias of the Eurasian steppes has switched from positive in JSBACHv3 to negative in JSBACHv4. This
difference is mainly caused by weaker precipitation and FAPAR of ICON-A+JSBACHv4 in July, which is most probably caused by a feedback loop between too little soil moisture, evaporation and clouds. While the size and patterns of biases in albedo and LST are largely similar between the two model versions, they are less well correlated for precipitation and vegetation related variables like FAPAR. Overall, we see very good perspectives for further improving JSBACHv4, because there are sound possibilities to mitigate most of the biases without interdependencies or technical problems.



# 1  Introduction

With the massively increasing parallelism in high performance computing during the last two decades (TOP500 project, 2021) unstructured grids became a favorable choice for general circulation models of the atmosphere and the ocean. Their usage offers good scalability, prevention of grid singularities at the poles, exact conservation of mass, and flexibility in resolution (e.g. for local refinements). Accordingly, the utilization of unstructured grids was the primary motivation to build the icosahedral nonhydrostatic (ICON) modeling framework for a unified next-generation numerical weather prediction and climate modeling system, a joint development of the German Weather Service (DWD) and the Max Planck Institute for Meteorology (MPI-M).

ICON was successfully introduced into the operational forecast system of the DWD in 2015 (Zängl et al., 2015; DWD, 2014). The atmosphere model ICON-A (Giorgetta et al., 2018) as well as the ocean model ICON-O (Korn, 2017; Korn and Linardakis, 2018) were established for climate simulations. Furthermore, a complex Earth System model named ICON-ESM is currently assembled based on ICON and a coupled version of ICON-A and ICON-O (Jungclaus et al., 2022).

As part of this major effort, the ICON-Land framework was developed at MPI-M to facilitate the implementation of complex land surface models not only in ICON-ESM, but also into other modeling environments for global simulations. ICON-Land has a code structure following object-oriented programming concepts. It offers management of processes and a hierarchical handling of tiles that represent different types of land cover within a grid box. As a first application of this framework, it now hosts JSBACHv4. As a consequence JSBACHv4 is the land component of ICON-A and thus also integral part of the first version of ICON-ESM. Technically, JSBACHv4 is a subroutine of ICON-A and part of its scheme for the implicit solution of vertical diffusion. Because of this intimate connection, the assessment of JSBACHv4 performed in the present study is done by means of *coupled* atmosphere-land simulations.

The base version of ICON-ESM – of which JSBACHv4 is part – is described and evaluated in a recent study by Jungclaus et al. (2022). Their evaluation focuses on the characterization of the main features of the new ICON-ESM as well as its performance in simulating the Earth system in its historical development. Our paper is a companion study to amend this evaluation by a more detailed investigation of the ability of its land component JSBACHv4 to represent land surface processes for climate modeling as well as for Earth System Modeling. For this purpose we focus on variables that are on the one hand meaningful for the physical and biological aspects of the overall model result and on the other hand are available as an observation-based dataset for comparison. For these reasons we selected albedo, Land Surface Temperature (LST), Terrestrial Water Storage (TWS), Leaf Area Index (LAI), Fraction of Absorbed Photosynthetic Active Radiation (FAPAR), Net Primary Production (NPP) and Water-Use-Efficiency (WUE) for our assessment. These variables represent processes that are fast compared to the land carbon cycle or climate induced biogeographical changes in landcover. Modules of JSBACHv4 representing the latter two are switched off in our simulations and are thus not part of the assessment.

JSBACHv4 is the successor of JSBACHv3 (Brovkin et al., 2013; Reick et al., 2013; Schneck et al., 2013; Reick et al., 2021), the land component of the Max Planck Earth System Model MPI-ESM 1.2 (Mauritsen et al., 2019) that participated in the Coupled Model Intercomparison Project (CMIP) phase 5 (Giorgetta et al., 2013) and phase 6 (Tebaldi et al., 2021). Most of the





parametrizations of JSBACHv4 are re-implementations from JSBACHv3. One aim of the present study is to check if the new

ICON-Land framework encompassing the implementation of JSBACHv4 is free from defects. At the same time we want to document the changes in the quality of simulation results when stepping from ECHAM6+JSBACHv3 to ICON-A+JSBACHv4. This is achieved by performing the same assessment as done for JSBACHv4 also for JSBACHv3 so that simulation biases can be compared. Hence, the present study is in fact a parallel assessment of the two JSBACH versions although our main interest lies in JSBACHv4.

We base our assessment on simulations with atmosphere coupling performed in AMIP configuration (Atmospheric Model Intercomparison Project; Gates 1992; Taylor et al. 2000). For JSBACHv4 we perform AMIP simulations of the last three decades with ICON-A that hosts JSBACHv4. For JSBACHv3 we use an existing AMIP simulation that was performed for the same period with the atmosphere model ECHAM6.3 that hosts JSBACHv3 as land component. The key characteristic of AMIP simulations is that the observed historical development of monthly sea surface temperature and sea-ice cover is

prescribed. Thereby most of the simulated interannual variability (e.g. related to the El Niño Southern Oscillation or the Pacific Decadal Oscillation) is in phase with the real climate. Moreover, such a model configuration prevents biases that arise in full ESM simulations from internal variability and biases in the ocean model; such biases would impede our discussion of land model performance. Besides, driving alternatively the considered land surface models by observed climate data in stand-alone mode would not permit their assessment as proper ESM components: such simulations would lack the interaction of the land

surface with the atmosphere and thereby presumably lead to smaller biases than those that have to be coped with in ESM simulations for which JSBACHv4 and JSBACHv3 were designed. For JSBACHv3, such biases arising from the coupling to atmosphere and ocean has been discussed for some variables in Dalmonech et al. (2015).

The structure of the paper is as follows: First we describe in section 2 our methodology by introducing the considered models and the simulation set-ups, then we describe each assessment variable, how it is calculated in JSBACH, by which

observational data it is assessed, and how observational data are pre-processed for this purpose. The subsequent section 3 contains our main results. There we compare for each assessment variable simulation results from the two JSBACH versions with observational data and investigate differences in simulation results from the two models. These results are discussed in section 4 with emphasis on the potential reasons for the emergence of simulation biases and their potential mitigation.

## 2 Methods

### 2.1 ICON-A and JSBACHv4

ICON-A is the atmospheric component from the Max Planck Institute for Meteorology (MPI-M) of the Icosahedral Non-hydrostatic Earth System Model (ICON-ESM) developed at the MPI and the German Meteorological Service (Deutscher Wetterdienst; DWD). ICON-A is described in Giorgetta et al. (2018) and evaluated in Crueger et al. (2018). It combines the nonhydrostatic dynamical core of Zängl et al. (2015) with the physical parameterizations adopted from the ECHAM6 GCM

(Giorgetta et al., 2013; Stevens et al., 2013). The dynamical core solves equations for the atmospheric motion, temperature, density, and concentrations of water in forms of vapor, clouds and cloud ice. The parameterizations of physical processes, like





radiation and cloud condensation, alter these dynamical variables. Crueger et al. (2018) found that, compared to ECHAM6.3, the representation of climate has slightly improved in ICON-A. For the present study we ran ICON-A in R2B4 resolution (grid spacing ∼160 km) and 47 layers for the atmosphere. The simulations covering the years 1979 to 2014 were performed in a
setup according to the AMIP II protocol (Taylor et al., 2000, 2012), prescribing monthly sea surface temperatures (SSTs), sea ice concentrations (Durack and Taylor, 2017), observed solar irradiance and historical greenhouse gas concentrations (Meinshausen et al., 2017).

The land component JSBACHv4 (Jena Scheme for Biosphere-Atmosphere Coupling in Hamburg) of ICON-A was developed by the MPI-M. JSBACHv4 is the successor of JSBACHv3 (Reick et al., 2013, 2021), but with a completely renewed structure.
However, it applies the same parametrizations as JSBACHv3 and includes the additional feature of frozen soil water and a five-layer snow scheme (Ekici et al., 2014; de Vrese et al., 2021). It comprises processes that are important for a land surface scheme: the surface energy balance, terrestrial water budget and runoff, surface exchange fluxes (moisture, heat, carbon), phenology, surface albedo and roughness, radiation in the canopy, plant productivity (photosynthesis, gross and net primary productivity), anthropogenic land cover change and land cover disturbances by wind or fire. In our simulation we use 11 plant
functional types (PFTs): Tropical broadleaf evergreen, Tropical broadleaf deciduous, Extra-tropical evergreen, Extra-tropical deciduous, Raingreen shrubs, Deciduous shrubs, C3 grass, C4 grass, C3 pasture, C4 pasture and C3 crops. Land cover change is prescribed by annual maps for the fractional cover of the PFTs within each grid cell. These maps are derived as described in (Mauritsen et al., 2019) from the Land Use Harmonization (LUH) project (Hurtt et al., 2011) version LUH2v2h (Hurtt et al., 2016, 2020) dataset. The radiation fluxes in the canopy that are needed in the photosynthesis routines to determine the Fraction
of Absorbed Photosynthetically Active Radiation (FAPAR) are calculated by employing a two-stream approximation (Sellers, 1985). The hydrological soil scheme uses five soil layers ranging 9.8 meter below the surface with increasing thickness towards the bottom. The model versions used in the present study are ICON-A of ICON-ESM-V1 and JSBACHv4.30.

## 2.2 ECHAM6 and JSBACHv3

To document the advancement from JSBACHv3 to JSBACHv4, we compare our ICON-A+JSBACHv4 results with data from
an existing AMIP simulation with JSBACHv3. This simulation was performed in the context of CMIP6 with the atmospheric component ECHAM-6.3.05 of MPI-ESM1.2-HR that hosts JSBACHv3.20 as land model (Mauritsen et al., 2019; Müller et al., 2018). The data are published as Max Planck Institute for Meteorology (2020). We used ensemble member r1i1p1f1. The JSBACHv3 AMIP setup is similar to our ICON-A run except for the model grid (T127 ∼ 100 km and 95 vertical levels), integer lake mask, no frozen soil, two snow layers, and the additional use of a C4 crop PFT (Table 1). In particular, it applies
the same land cover change scheme and input maps as our ICON-A experiment.

## 2.3 Assessed variables and their representation in JSBACH

Here we introduce the variables that we selected for our model assessment. Only variables describing fast processes (seconds to years) are considered. Slow processes (decades and longer, e.g. climate induced changes in biogeography and wood or soil





**Table 1.** Simulation differences between JSBACHv4 and JSBACHv3.

| Parametrization | JSBACHv4 | JSBACHv3 |
|---|---|---|
| Plant Functional Types | 11 | 11 + C4 crops |
| Frozen soil | yes | no |
| 5-layer snow scheme | yes | no |
| Fractional lakes | yes | no |

carbon turnover) are as well implemented in JSBACHv4 but are not subject to assessment here. As process descriptions are
largely similar in JSBACHv4 and JSBACHv3, in the following we distinguish between those two versions only when necessary.

### 2.3.1 Albedo

Land surface albedo controls the balance of shortwave radiation at the surface and thus the land energy uptake. It plays a role
for the land biosphere as well as for the climate of the lower atmosphere. Its changes are shown to have positive feedback
effects with climate (e.g. Claussen, 1997). Changes are driven by natural seasonal and diurnal alterations, as well as human
interventions like vegetation changes due to land use (Forster et al., 2007). Surface albedo depends on the canopy properties
(e.g. LAI), soil color, the color of litter covering the soil, as well as on snow cover and changes in snow color when aging. As all
these properties and the surface albedo itself are typically calculated by the land surface scheme (land model), the albedo is one
of the most common variables for its evaluation. This is especially true in consideration that albedo results vary strongly among
land models. For example, Levine and Boos (2017) show for northern hemisphere summer that intermodel albedo variance in
CMIP5 (Coupled Model Intercomparison Project Phase5) simulations is large compared to interannual albedo variance. Wang
et al. (2016) also show a strong intermodel albedo variability in the CMIP5 simulations but for northern hemisphere winter.

The albedo scheme of JSBACH computes temporal and spatial albedo changes. The scheme differentiates between near
infrared (NIR) and visible range (VIS) albedo, only for lakes this differentiation is not done. On lakes, the fraction of lake ice
and snow on lake ice is taken into account. The albedos for PFTs and bare soil as well as for snow are computed separately.
Snow albedo decreases with increasing snow age and surface temperature. The overall albedo on land is then calculated from
the fractions of soil, canopy and the corresponding overlaying snow, considering the influence of the incoming solar radiation
angle on their fractions. Thereby the coverage of the soil by stems and branches within forest is considered. JSBACHv4 uses
the same albedo scheme as JSBACHv3 which is described in detail in Otto et al. (2011).

As a benchmark for our assessment of the JSBACH albedo results we use the MODIS MCD43C3 CMG Collection 6 Albedo
Product (Schaaf and Wang, 2015) which is suitable for climate model comparisons (Cescatti et al., 2012; Schaaf et al., 2002).
For our comparison we exclude data flagged for minor quality from inversion (quality flag 4 and 5).





The contributions from snow cover make a large contribution to the albedo values in the extra tropics. Therefore, as part of the discussion of albedo results, we also discuss the simulated distribution of snow cover and compare it with observed snow cover from the same MODIS dataset.

### 2.3.2 Land Surface Temperatures (LST)

Virtually all land processes depend directly or indirectly on LST. It is a key variable in the surface energy balance and thus takes part in the control of thermal, radiative, and hydrological exchange fluxes at the interface between land and atmosphere that shape local climate and the state of the lower atmosphere. It plays a central role for cryospheric processes (amount and duration of snow cover, formation of soil ice) and determines local living conditions for fauna and flora.

Atmospheric temperature is one of the most regarded prognostic variables in climate models. Its calculation depends over land on LST as lower boundary condition that is provided by the respective land component. In JSBACH land surface temperature is obtained from the surface energy balance equation, whose solution is embedded in the vertical diffusion scheme for heat and moisture fluxes in the atmosphere (implicit coupling). LST is also used as upper boundary condition to determine the vertical temperature profile within the five-layer soil model assuming vanishing heat fluxes at the bottom (10-m depth).

For comparison of simulated LST with observations, we use the MOD11C1 Moderate Resolution Imaging Spectroradiometer (MODIS) Terra Land Surface Temperature/Emissivity V006 dataset (Wan et al., 2015). We excluded data points where the quality flags indicate no retrieval because of clouds.

As a part of the LST analysis we regard the modelled Total Cloud Cover (TCC) as compared to observed TCC from the Collection 6.1 EOS-TERRA MODIS Atmosphere Level-3 Daily Product (Platnick, 2017).

### 2.3.3 Terrestrial Water Storage (TWS)

TWS influences many surface properties connected to climate. Especially its particular representation in land surface schemes has a major impact on calculated evaporation, transpiration (root water uptake), NPP and LAI (water limitation). As a result, the latent heat flux into the atmosphere heavily depends on TWS. On a global scale, its seasonality peaks in about April and is lowest around September (Swenson and Milly, 2006). Due to the local seasonality of precipitation and evapotranspiration, TWS has a strongly site dependent seasonality (Feng et al., 2012; Hickel and Zhang, 2006; Settin et al., 2007). Even when TWS has a strong dependency on the atmosphere through precipitation and air temperature it is still determined by surface properties like the soil type and vegetation. Therefore, its depiction in climate models depends largely on the particular land model. Koster et al. (2009) drove a number of land surface models with the same atmospheric forcing and concluded that soil moisture is highly dependent on the particular land surface model.

TWS encompasses water in snow, vegetation, soil, runoff and aquifers. In JSBACH, TWS is the sum of water stored in snow at the surface and at the canopy, soil and soil ice, runoff and in the skin reservoir on the surface. Aquifer water impacts climate only when it emerges as or in open water (runoff, lake or ocean). Therefore, aquifer water is not explicitly represented in JSBACH (as in most land models). JSBACH also has no explicit store for water in vegetation. Formally, this water is part of the soil water pool because for transpiration the water is taken directly from the soil water pool. The water budget in JSBACH



is calculated from processes above ground, the soil hydrology and the river runoff. Above ground processes are the snow and water that fall on the surface or that are intercepted by the canopy. The water that reaches the surface (as rain, snow melt and dew) can infiltrate the soil. From that only a part is taken up by the soil, the remaining part enters the surface runoff. Vertical movements of water in the soil are the result of its vertical diffusion and its gravitational percolation. Water in- and output occur at the surface in form of rain, evaporation and snow melt (on the surface and the canopies). Water transpired by the vegetation

is extracted from the soil water reservoir. The size of transpiration depends on primary production, root depth, and specific humidity of the lowest atmospheric layer. At the bottom of the soil water is lost as drainage which is added to the overall runoff. Because TWS could not be calculated from the ECHAM6+JSBACHv3 simulation output we only assess JSBACHv4.

The Gravity Recovery And Climate Experiment (GRACE) is a project to accurately determine the Earth's gravitational field with detailed measurements from satellites. For our assessment of simulated TWS we use the ITSA_Grace2018s unconstrained

monthly TWS Dataset (Kvas et al., 2019; Mayer-Gürr et al., 2018). It is derived from the GRACE satellite sensing of changes in the gravitational field of the Earth. These changes are reprocessed to reduce the effects of mass trends originating from glacial isostatic adjustment, postseismic deformation after large earthquakes and atmospheric mass variability. Therefore, the dataset essentially senses water mass anomalies independently of their surface exposure and thus integrates all water mass changes in snow cover, vegetation, soil, runoff and aquifers. For our assessment we are interested in the changes of TWS in

the course of the year. As model results and GRACE data are derived by totally different methods we expect that the size of monthly TWS change will systematically differ. For example, both data sets are expected to have different standard deviations. Before comparison we therefore normalize the observed and simulated data according to their own size of variation. For the normalization we use averaged absolute values instead of the standard deviation. Using the standard deviation would put more weight on extreme values, which makes sense to lower the impact of statistical outliers. However, here statistical outliers are

unlikely as we regard multi year averages. Therefore we use for each particular grid cell:

$$\overline{\Delta x} = \frac{1}{12} \sum_{m=1}^{12} |\overline{x}_m|,$$

where $\overline{x}_m$ is the average of observed TWS change for month $m$ calculated over the period of available observation data (years 2003 to 2014). From this we calculate the normalized month-to-month differences as

$$\Delta \text{TWS}_m := \frac{\overline{x}_{m+1} - \overline{x}_m}{\overline{\Delta x}},$$

with m = 1, 2,...11. Denoting simulated values by TWS$'$ and applying to them the same calculations, we measure the average mismatch between simulated and observed TWS in month-to-month changes across a year by

$$Q_{\text{TWS}} := \sum_{m=1}^{11} |\Delta \text{TWS}'_m - \Delta \text{TWS}_m|.$$

This value can reach a maximum of 22 and the closer to zero it is, the more synchronous are observed and simulated month-to-month changes.





### 2.3.4 Leaf Area Index (LAI)


The LAI strongly affects the exchange of energy and matter with the atmosphere through its impact on albedo and the fluxes of water and carbon (Chase et al., 1996; Betts et al., 1997; Piao et al., 2007). A high LAI typically reduces albedo and enhances transpiration of water and in general also primary productivity. The LAI depends primarily on the climatic conditions for a given biome. With the onset and eventual loss of leaves at the beginning and end of the growing season the LAI reveals a strong

seasonality. However, in contrast to soil moisture, it remains unaffected by short precipitation spells. In climate models, the LAI is time-dependent and either prescribed from observations or calculated by the corresponding land surface scheme. How it is calculated strongly depends on the particular land model (see e.g. Wang et al., 2016).

JSBACHv3 provides different schemes to compute the LAI (Reick et al., 2021). The default scheme, which is also implemented in JSBACHv4, is the LoGro-P model (Logistic Growth Phenology). Each PFT is assigned to one of the following

phenology types: evergreen, summergreen, raingreen, grasses, tropical and extra-tropical crops. Each phenology type has a maximum LAI, which represents its physiological limit. Changes in the summergreen phenology are based on three phases: a growth (in spring), vegetative (in summer) and a rest phase (in autumn and winter). The raingreen, grasses and tropical crop phenology types only have a growth phase; they grow whenever the environmental conditions (soil moisture, temperature, NPP) are favorable. The evergreen and extratropical crop phenologies have no vegetative phase. Dependent on the phenology

type, different environmental conditions determine the advancement from one phase to the next. For example, the evergreen, summergreen and extratropical crop phenologies use accumulated temperatures (heat sum) for entering the growth phase when it reaches a PFT-specific threshold value. Overall, the LAI increases when the soil is wet and temperatures and NPP are high. A detailed description of the LoGro-P model is given in Böttcher et al. (2016).

For the assessment, we compare our simulation results for LAI with the MODIS MOD15A2Hv006 LAI Product (Myneni

et al., 2015). We use only data where the primary or secondary quality flag reveal more than a 50% fraction of measurements with best or good quality (primary/secondary quality flag of 51-100 and 251-300).

### 2.3.5 Fraction of Absorbed Photosynthetic Active Radiation (FAPAR)

The Fraction of Absorbed Photosynthetically Active Radiation (FAPAR) quantifies the fraction of the Photosynthetically Active Radiation (PAR, solar radiation in the 400 -700 nm spectral domain) absorbed by leaves for photosynthesis. FAPAR depends

on the LAI, the optical properties of the leafs and their orientation, atmospheric conditions, the angle of incoming radiation, and the albedo of the underlying soil. It determines Gross Primary Productivity (GPP) and plays a key role for plant respiration and transpiration.

To solve the complicated radiation problem JSBACH uses a so called two stream approximation approach. This approximation requires that the leaves are distributed homogeneously in the canopy and the radiation distribution within the canopy is

horizontally invariant. Hence for closed canopies it is sufficient within the canopy to consider vertically up- and downward radiation fluxes. However, the concrete implementation of this approach into JSBACH is rather complex and we refer the reader to Loew et al. (2014) for details.





For our assessment of FAPAR we regard the differences between our model and the MODIS MOD15A2Hv006 Product. We took only the data where the primary or secondary quality flag have more than a 50% fraction with best or good quality

measurements.

### 2.3.6 Net Primary Production (NPP)

The exchanges of carbon between terrestrial ecosystems and the atmosphere are dominated by GPP and plant respiration (Houghton, 2007). NPP, as the net carbon flux from these two, is therefore very important for the carbon balance between atmosphere and land. It depends on LAI, TWS and WUE as well as on temperature, radiation and fire. It varies seasonally

and annually, depending on the climatic conditions (precipitation, temperature and radiation). Globally it peaks in northern hemisphere summer and has its low point in winter. As shown in Cramer et al. (1999) its representation strongly depends on the land model.

In JSBACH GPP is calculated from FAPAR under consideration of water stress. Water stress is quantified by calculating the relative amount of water found in the root zone taking into account the wilting point below which plants cannot extract any

more water from the soils and above which transpiration is largely unhindered. In JSBACH carbon is respired on the one hand as maintenance respiration (to cover the basic plant functions like photosynthesis, nutrient and water transport, repairs, defence) and on the other hand as growth respiration (to cover plant growth). In JSBACH the net balance of GPP and respired carbon is called potential NPP. In times when more carbon is respired than gained from the assimilation potential NPP is negative. In this case as much carbon as possible is taken from the reserve pool (representing carbon in plants stored in sugars and starches) to

obtain the actual NPP. If potential NPP is positive and can be allocated to the plant carbon pools or through root exudates to the soil carbon, potential NPP equals actual NPP. However, if potential NPP can not be allocated because of structural limits of the plants (plants can not grow infinitely large) as much carbon as possible is allocated (actual NPP is smaller than the potential NPP) and the remaining carbon is added as root exudates to the soil carbon.

We use the MODIS-C006_MOD17A2Hv006 dataset as benchmark for our actual NPP assessment (Running and Zhao,

2019). We use only data with confidence flag having the values 0 (very best possible) and 1 (good, very usable, but not the best).

### 2.3.7 Water-Use-Efficiency (WUE)

Continuous water uptake is a prerequisite for plant growth. But how much they need to photosynthesize a unit of organic carbon is different between different plants and, at a larger scale, different between ecosystems. This efficiency by which

plants use water is closely related to the way plants operate their leaf stomata: By closing them less leaf water evaporates into the surrounding air which reduces the risk of desiccation, while their opening facilitate access to atmospheric $CO_2$ and thereby assimilation and growth. In dry areas growth of plants is typically limited by lack of water and they have developed means to use water to assimilate carbon more efficiently than plants in wet regions (e.g. $C_4$ instead of $C_3$ carbon fixation). But even under the same climate conditions Water-Use-Efficiency (WUE) can differ depending on the species (Niu et al., 2011), stand





age and ecosystem structure. Overall, WUE strongly defines carbon and water exchanges with the atmosphere (Sun et al., 2016).

WUE can be characterized in different ways (Beer et al., 2009). In the context of land surface models with their large grid cells only a definition at biome or ecosystem level is useful. Here we define WUE as the ratio between annual GPP and annual evapotranspiration (ET) integrated over all PFTs in a grid cell. This characterizes WUE at ecosystem level as it employs

evapotranspiration that not only includes transpiration but also evaporation from ground. Moreover, this definition has the advantage that for GPP and ET global observational data sets exist from which this WUE can be calculated.

In JSBACH, GPP is derived in a two step approach. First, *potential* GPP is calculated assuming absence of water stress so that the plants photosynthesize at maximum rate. Then the soil hydrology model computes the corresponding potential water loss from transpiration. In a second step *actual* GPP is calculated accounting for a possible lack of soil water. JSBACH

differentiates between C3 and C4-photosynthesis. For C3-plants an implementation of the Farquhar et al. (1980) model and for C4-plants an implementartion of the Collatz et al. (1992) model is used. Both models take into account that the carboxylation and electron transport rate depend on temperature.

We calculate the JSBACH WUE for each grid cell as total actual GPP from all PFTs divided by total evapotranspiration in that grid cell and average these quotients over the years 2001 to 2014. We compare the JSBACH results with the WUE

calculated from MODIS data. For that we divide the MODIS-C006_MOD17A2Hv006 GPP (Running and Zhao, 2019) with MODIS MOD16A3GF.061 evapotranspiration (Running and Zhao, 2019). For evapotranspiration we excluded data flagged with a minor quality above 80% (bad days/total days). We compare the JSBACH results also with the global distribution of WUE published by Sun et al. (2016) calculated partially from different observation based data. To ease visual comparison, we plotted our figures such that contour levels and colors are as theirs.

## 2.4  Data Preparation

Table 2 summarizes the variables that are compared with corresponding observational data. The spatial resolution of the MODIS data is 0.05° x 0.05° (about 5.6 km at the equator), of the GRACE data 2° x 2° (about 220 km at the equator), and of the GPCP data 2.5° x 2.5° (about 280 km at the equator). The observational data as well as our model results are interpolated with a first order conservative remapping to a Gaussian 96 x 192 lonlat grid (T63; about 1.88°/210 km at the equator).

In the MODIS data quality flags are included which indicate issues from elicitation arising from atmospheric scattering and absorption, anisotropy, inadequate temporal, spatial and spectral sampling, and narrowband to broadband conversions. For the MODIS data the interpolation is done after the data points with minor quality are excluded (see subsections above for details). Except of TWS, we average our model results, the MODIS data, and the GPCP data over the years 2001 to 2014. The TWS and the corresponding model data are averaged over the years 2003 to 2014 because the GRACE dataset misses 2001 and parts of

2002. For our precipitation results we use only grid cells where the differences between JSBACH and GPCP are lager than the GPCP error bar. For the assessment of the seasonality we selected January and July to represent deep winter and high summer in the two hemispheres. The respective maps were obtained by averaging all Januarys and Julys in the considered time range 2001 to 2014. All presented spatial correlations are weighted with the grid cell area.





**Table 2.** Variables and the observational data used for their assessment.

| ICON-A+JSBACHv4 variable | Corresponding observational data | Reference |
|---|---|---|
| Albedo and snow cover | MODIS MCD43C3 CMG Collection 6 Albedo Product | Schaaf and Wang (2015) |
| LST | MODIS MOD11C1 Terra Land Surface Temperature/Emissivity V006 | Wan et al. (2015) |
| TCC | Collection 6.1 EOS-TERRA MODIS Atmosphere Level-3 Daily Product | Platnick (2017) |
| Precipitation | GPCP monthly precipitation dataset from 1979-2021 | Adler et al. (2003) |
| TWS | ITSA_Grace2018s unconstrained monthly TWS Dataset | Mayer-Gürr et al. (2018) |
| LAI | MODIS MOD15A2Hv006 LAI Product | Myneni et al. (2015) |
| WUE | see GPP and Evapotranspiration; Sun et al. (2016) | Sun et al. (2016) |
| FAPAR | MODIS MOD15A2Hv006 | Myneni et al. (2015) |
| NPP | MODIS-C006_MOD17A2Hv006 | Running and Zhao (2019) |
| GPP | MODIS-C006_MOD17A2Hv006 | Running and Zhao (2019) |
| Evapotranspiration | MODIS MOD16A3GF.061 | Running et al. (2021) |

Abbreviations: Land Surface Temperature (LST), Total Cloud Cover (TCC), Terrestrial Water Storage (TWS), Leaf Area Index (LAI), Water-Use-Efficiency (WUE), Fraction of Absorbed Photosynthetic Active Radiation (FAPAR), Net Primary Production (NPP), MODerate resolution Imaging Spectroradiometer (MODIS), Gravity Recovery And Climate Experiment (GRACE), Global Precipitation Climatology Project (GPCP).

## 3 Results

Here we present our results from ICON-A+JSBACHv4 and ECHAM6+JSBACHv3 simulations. One general tendency is that simulation results from the two models are quite similar, so that often the discussion of the results applies to both models – in such cases we simply talk of 'JSBACH' without version number. In all other cases, when we discuss specifics of results from a particular model version, the full model name will be used. Because biospheric variables strongly depend on environmental conditions, we discuss physical variables first.

### 3.1 Albedo

**Table 3.** Spatial Spearman rank correlations (rho) between the biases of JSBACHv4 against those of JSBACHv3 for all grid cells.

| | Albedo VIS | Albedo NIR | LST | LAI | FAPAR | NPP |
|---|---|---|---|---|---|---|
| Year | - | - | 0.854 | 0.658 | 0.336 | 0.735 |
| January | 0.754 | 0.756 | 0.856 | 0.735 | 0.333 | 0.769 |
| July | 0.844 | 0.829 | 0.701 | 0.599 | 0.340 | 0.551 |

Fig. 1 shows for both JSBACH versions the albedo biases, separately for the visible (VIS) and near infra red (NIR) range. The bias pattern is very similar for JSBACHv3 and JSBACHv4 with exceptions only found in central North America and some



**Figure 1.** Bias in White Sky Albedo (WSA) for JSBACHv4 (left) and JSBACHv3 (right) compared to MODIS data averaged over the years 2001 to 2014. Shown are biases in the near infrared (NIR) and visible (VIS) range for January and July. Because of polar night no observation data are available for arctic and antarctic winter. Note that significance is not shown because nearly all differences are significant (p=0.05) according to an independent two-sample t-test.





regions in North-Eastern Europe (e.g. in the January values of the VIS albedo). The similarity of bias patterns between the two model versions is visible in Table 3. It is also visible from the scatter plots in Fig. A2: The correlations (r and rho) are about

0.7 and higher except of r∼0.6 for VIS albedo in January when a larger scatter arises from polar regions.

For the biases seen in Fig. 1 three main causes can be identified. First, the too low VIS albedo and the too high NIR albedo over glaciers (Antarctica, Greenland) is likely a direct result of the fixed minimum and maximum albedo values used in the JSBACH calculations of glacier albedo. Second, the strong negative VIS and NIR albedo bias seen for January in western N-America, eastern Europe and large parts of central Asia is most probably a result of the bias in the dynamically calculated

snow cover fraction (compare Fig. 2). And third, the biases in all other regions that are not covered by glacier or snow are at least partly caused by old canopy and soil albedo maps used in JSBACH. These biases will be explained in the next paragraph.

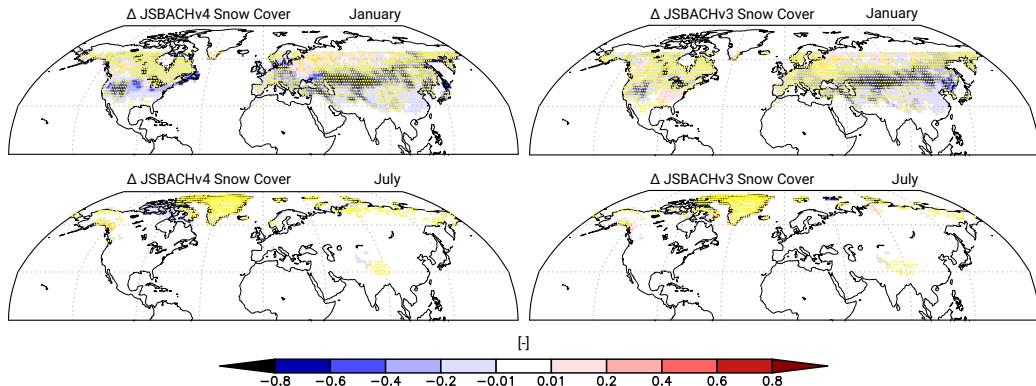

**Figure 2.** Bias in snow cover for JSBACHv4 (left) and JSBACHv3 (right) compared to MODIS data. Shown are the averages of January (top row) and July (bottom row) averaged over the years 2001 to 2014. The yellow dots indicate grid boxes where the bias is significant (p=0.05) according to an independent two-sample t-test.

Our analyses reveals that the reason for the too low NIR albedo obtained for northern midlatitudes summer is the static map used for soil albedo. The origin of the mostly too large albedo values in other regions (S-America, Africa, India, Australia) are not traceable to single causes but must be a geographically complex result of the calculated superposition of the assumed

values for canopy albedo (which depend on vegetation type) and the static map for soil albedo. A closer inspection (not shown) indicates that in the NIR range JSBACH has a too high canopy and a too low soil albedo while for VIS albedo the converse must be suspected. JSBACH uses the same canopy and soil albedo maps for the whole year and does not incorporate seasonality. The canopy and soil albedo maps used in JSBACH are derived from a fit (see discussion for details) to the 2001 to 2004 mean of an older MODIS Product (MODIS MOD43C1 CMG Collection 4 Albedo Product, Strahler et al. (2021), Gao et al. 2005).

As the version used internal in JSBACH is a yearly mean one would expect strong differences to observations for January and July. In Fig. 3 we show the difference between the version used internal in JSBACH and the MODIS observations for January and July we use here for our bias analysis. These differences explain a considerable part of the albedo biases seen in Fig. 1: In Africa for example for both months considered and both spectral ranges analyzed large parts of the JSBACH biases are already visible as bias of the older compared to the newer MODIS product. The same holds for South-East Asia, and for the





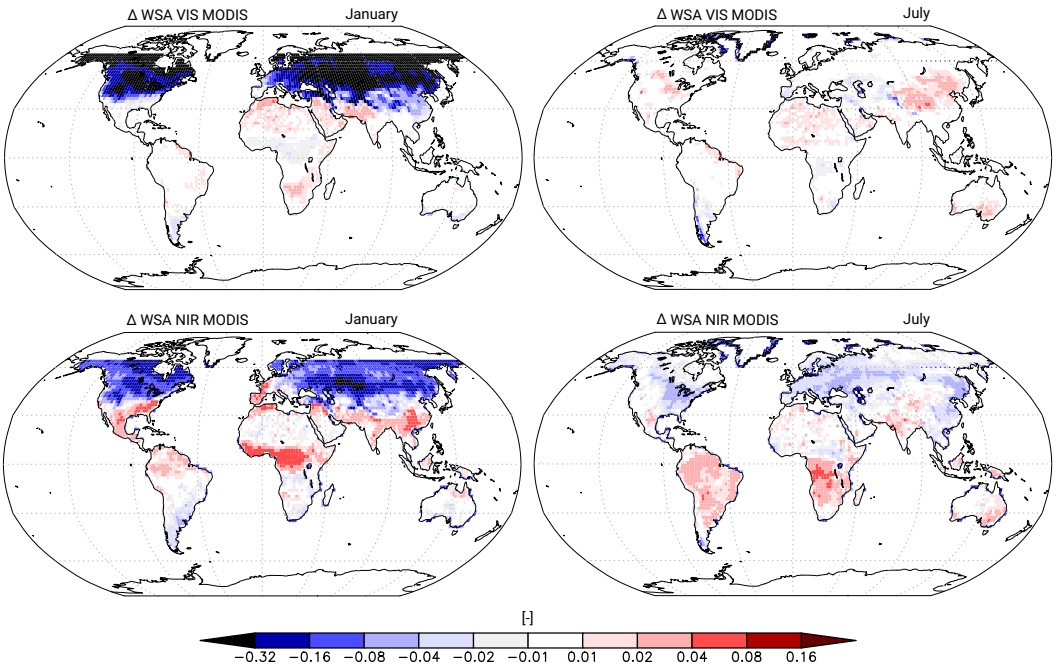

**Figure 3.** Difference in White Sky Albedo (WSA) between the two versions of MODIS data averaged over the years 2001 to 2014: The older one, from which the soil albedo maps used in JSBACH were derived, and the newer one used for our bias analysis in Fig. 1. Shown are NIR and VIS. As JSBACH does not account for seasonality in the soil albedo maps, the old version is a yearly mean. From this version we subtracted the January and July values of the newer version. The differences shown here can be interpreted as the 'bias' of JSBACH compared to the observations for January and July.

NIR albedo in India. Accordingly, a considerable part of the biases in albedo arise from the maps for canopy and soil albedo. Note that the rather large differences between the two albedo products seen in Fig. 3 at high latitudes in northern winter in the visible range are irrelevant for our analysis here, because in these regions at that time of the year the values from the static maps of soil and canopy albedo do not enter the calculated albedo value in JSBACH because of snow there.

### 3.2 Land Surface Temperature (LST)

Global mean LST of 9.6° C simulated by JSBACHv4 matches well with observational data (see Fig. 4 top). The lower northern hemisphere winter values found for JSBACHv4 (Fig. 4 bottom) are the main reason for the overall about one degree colder climate compared to JSBACHv3 and the better fit with the MODIS data. For JSBACHv4 the values for January are up to two degrees lower than for JSBACHv3, otherwise the seasonality is very similar for the two model versions and the phasing is consistent with MODIS observations.

In Fig. 5 we show the geographic distribution of temperature biases with respect to MODIS; to the left we show the biases for JSBACHv4, and to the right those for JSBACHv3. For both models the strong regional biases cancel each other in the global



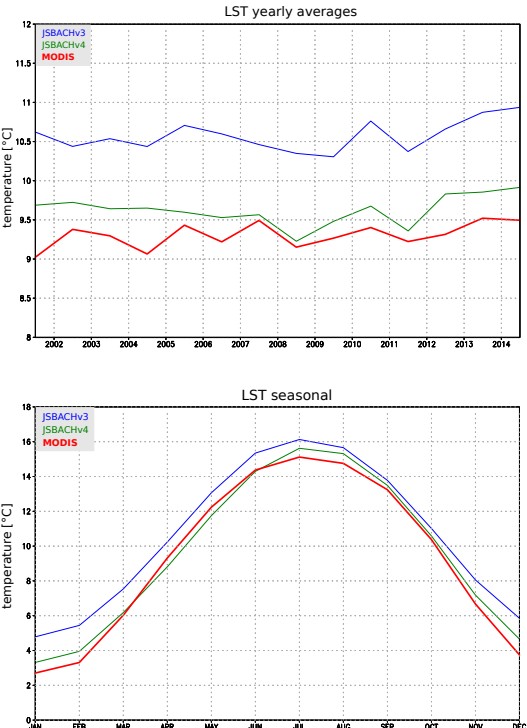

**Figure 4.** Global mean LST for JSBACHv3, JSBACHv4, and observation values from MODIS. Top: Yearly averages from 2001 to 2014. The vertical dashed lines indicate January 15th of the respective year. Bottom: Monthly mean values over the time period form 2001 to 2014. The vertical dashed lines indicate the 15th of the respective month.

mean leading to the relatively weak biases in Fig. 4 (top). At annual mean (Fig. 5 top row plots), the geographic distribution of temperature biases is very similar for the two models, and their difference is typically smaller than the bias. The zonal plots to the left and right reveal that the higher northern latitudes warm bias is as well significant when considered as a large

scale phenomenon (for regions north of 45°N the uncertainty range seen in the zonal plots lies off the line of zero bias). This temperate to boreal warm bias is for both model versions visible during northern winter (see the mid row plots for January), but less strong for JSBACHv4. In July the bias is much more pronounced in JSBACHv4 (mainly around 50°N, also visible in Fig. A3) and in contrast to JSBACHv3 significant in the zonal mean (see Fig. 5). In January the above named snow-albedo feedback surely contributes to the warm bias in JSBACH. In July the negatively biased JSBACHv4 precipitation (Fig. 7)

obviously supports the warm bias in Eurasia through the associated evaporation reduction (Table A1 shows a correlation of ∼-0.6). Table 4 also shows a high correlation (∼-0.8) between the JSBACHv4 July LST bias in Eurasia and a weaker Total Cloud Cover (TCC, see Fig. 6). It seem obvious that the JSBACHv4 biases in this area are the result of a feedback loop: too little clouds and precipitation result in too low RSM and evaporation which feeds back to precipitation and is associated with the too high LST. In the follwing this will be called the precipitation-RSM feedback. It is probably caused by the frozen soil and

5-layer snow scheme implemented in JSBACHv4 (see discussion). In contrast, in JSBACHv3 the precipitation-RSM feedback seems to have no importance in this region. Instead the January warming effect of the TCC is more pronounced in JSBACHv3



**Figure 5.** Bias in Land Surface Temperature (LST) compared to MODIS data averaged over the years 2001 to 2014. Shown are the annual averages (top row), and averages of Januaries (mid row), and Julys (bottom row). The zonal and spatial plots in the left two columns refer to JSBACHv4, while those in the right two columns refer to JSBACHv3. The grey range in the zonal plots indicates the 95% uncertainty range calculated from a Student t-distribution so that zonal values are significantly different when the zero bias line is found outside this grey range. Note that the significance for the grid cells of an independent two-sample t-test is not shown because nearly all biases are worldwide significant (p=0.05).

(see Table 4). However, for both models a higher TCC has a warming effect in January - as expected from the general rule that a cloud cover tends to cool the surface in the summer hemisphere and tends to warm the surface in the winter hemisphere (Chen et al., 2000).



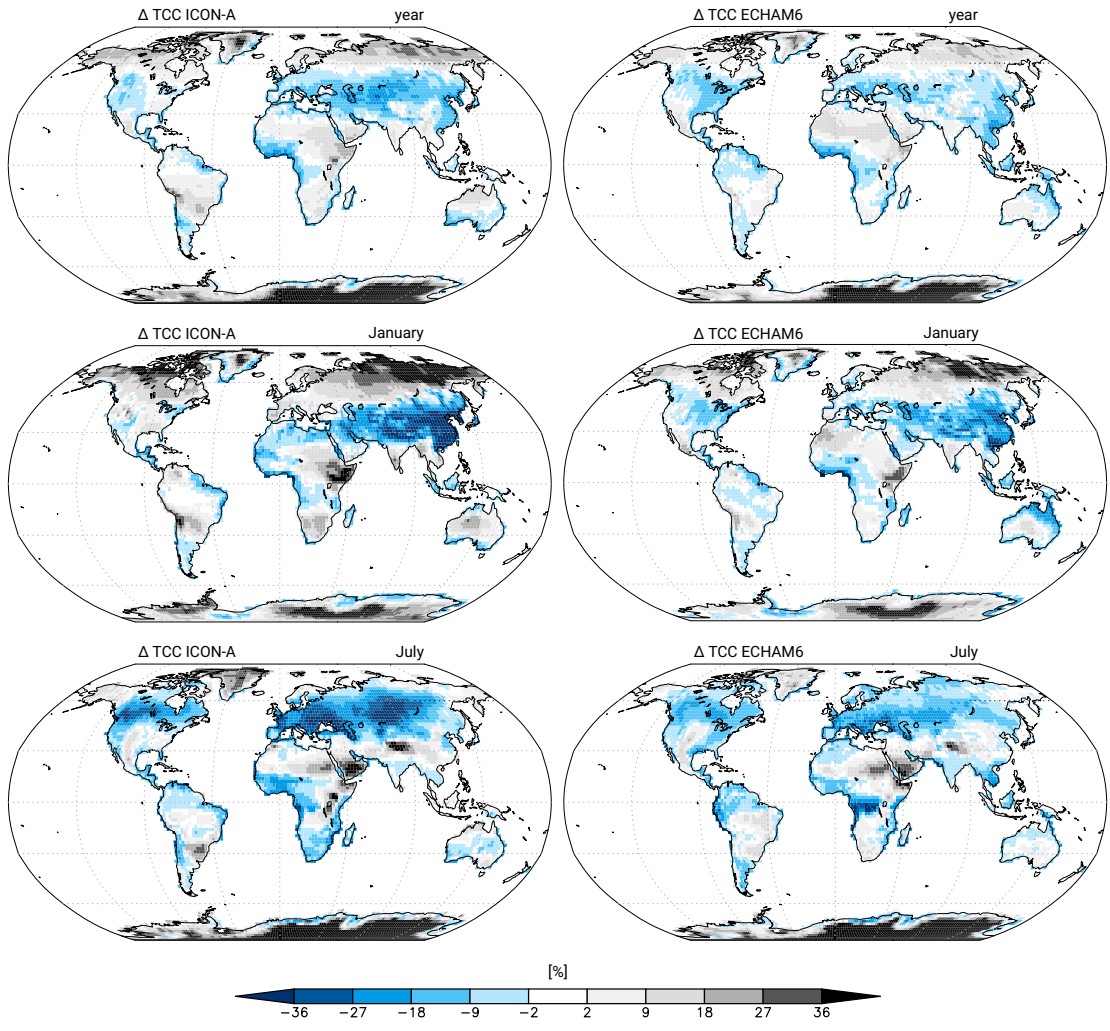

**Figure 6.** Bias in Total Cloud Cover (TCC) for ICON-A (JSBACHv4) and ECHAM6 (JSBACHv3) compared to MODIS data. Shown are the annual, January and July means over the years 2001 to 2014.

The Antarctic and Greenland ice sheets tend to be warm biased. Only for JSBACHv4 in January Greenland shows no uniform pattern. Cold biases are seen in northern and southern subtropics, and are more expressed for JSBACHv4. These two observations together – lower high latitudes warm bias, stronger subtropical cold bias – are together the reason for the globally lower temperature seen for JSBACHv4 in the plot of Fig. 4 (top). This is somewhat more pronounced for January which is visible in the zonal averages and in Fig. 4 (bottom).

The differences in bias patterns between the two model versions are also visible in the scatter plots of Fig. A3: The low July correlation (r=0.653) is caused by JSBACHv4 being warmer in the 'temperate zone' and colder in the 'polar zone' (for the





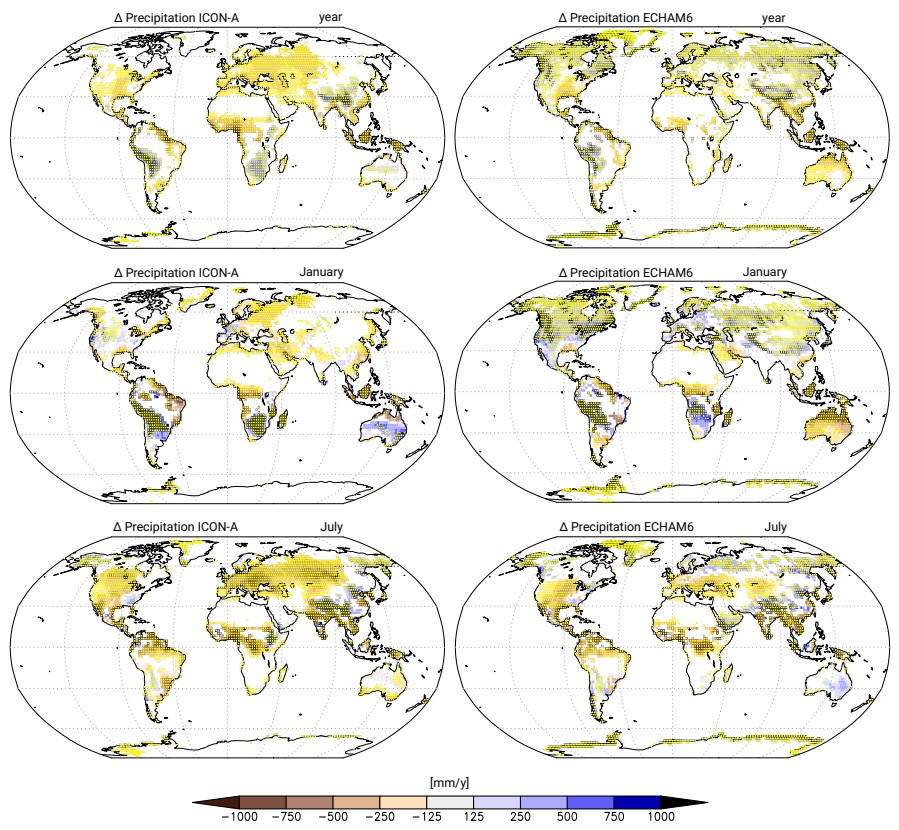

**Figure 7.** Bias in precipitation for JSBACHv4 (left) and JSBACHv3 (right) compared to GPCP precipitation. Shown are the annual, January and July means over the years 2001 to 2014. Note, land areas where the bias is smaller than the GPCP error bar are not plotted. The yellow dots indicate grid boxes where the bias issignificant (p=0.05) according to an independent two-sample t-test.

definition of the zones see Fig. A1). For January and also for the whole year, LSTs are much stronger correlated (r>0.84). This is also true when we regard the corresponding rank correlations of Spearman (Table 3).

Overall, for both models the bias pattern is very zonal, showing a warm bias at northern and southern higher latitudes, a cold
bias in the northern and southern subtropics, and an insignificant bias in the equatorial regions (5). The north-south symmetry of this zonal bias pattern hints to an atmospheric origin in the simulated climate since for biases caused by land processes one would not expect a north-south symmetry because of the very different continental distribution in the two hemispheres.

In summary we find that: (1) The global mean LSTs of JSBACH only fit the MODIS data because regional biases cancel each other. The global mean LSTs of JSBACHv4 fit better with the MODIS data because of its lower January values in the
subtropics. (2) Because of the overall zonal bias pattern of both models a strong atmospheric contribution has to be assumed. (3) Both models exhibit a higher northern latitudes warm bias. In January, especially in JSBACHv3, the TCC partly causes this warming and both models show a strong amplification by the snow-albedo feedback. In July, but only in JSBACHv4, a





feedback between precipitation and RSM causes a warm bias. (4) Except of Greenland in January for JSBACHv4 the Antarctic and Greenland ice sheets tend to be warm biased.

**Table 4.** Spatial Spearman rank correlations (rho) between the JSBACHv4 and JSBACHv3 biases of TCC against those of LST and FAPAR.

|  | JSBACHv4 | | JSBACHv3 | |
|---|---|---|---|---|
|  | TCC vs. LST (Eurasia) | TCC vs. FAPAR | TCC vs. LST (Eurasia) | TCC vs. FAPAR |
| Year | 0.027 | 0.217 | 0.596 | -0.016 |
| January | 0.141 | 0.218 | 0.593 | 0.134 |
| July | -0.797 | 0.138 | -0.329 | -0.060 |

Eurasia refers here to the square between longitude 0° to 184° and latitude 35° to 90°.

## 3.3 Terrestrial Water Storage (TWS)

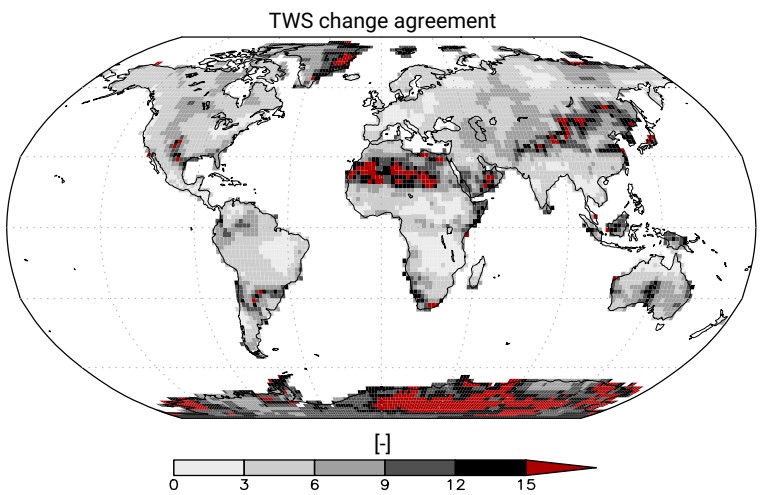

**Figure 8.** Normalized TWS change differences in the course of the year between the JSBACHv4 and GRACE data. Note that, due to the statistics, only the relative agreement between different regions can be rated but not an amount of water mass change agreement in absolute terms. However, the higher the value the stronger is the difference in the course of the year between JSBACHv4 and GRACE.

Fig. 8 shows the agreement between JSBACHv4 and GRACE in the alterations of TWS changes in the course of the year. The strongest differences occur in dry regions with small TWS changes. In these regions precipitation is very sensitive to the exact atmosphere dynamics and thus small shortcomings result in strong effects in Fig. 8. At the same time in regions where precipitation is not very sensitive to exact atmospheric conditions the model is in better agreement. For example, the inner





tropics with strong deep convection (central Africa, central S-America, northern Australia), regions that are strongly affected
by the monsoon (India, Africa) and regions that are dominated by the westerlies (Europe, and eastern Europe). Also north-
eastern Asia reveals relatively good agreements despite the fact that the amount of TWS change is not high there. In contrast
to this general rule, quite the opposite applies for the Indonesian islands. This is probably due to the resolution of the model
which can not depict precipitation very exactly for small land areas surrounded by ocean. We find the same problem also for
all extratropical islands.

### 3.4  Leaf Area Index (LAI)

In Fig. 9 we show the bias in LAI for JSBACHv4 and JSBACHv3 compared to MODIS LAI data. The bias patterns between
both models are strongly correlated (see Table 3). In the tropics the bias pattern is quite similar. Differences between the models
are mainly seen in the northern extratropics, where in July the LAI is too large for JSBACHv4 across the high northern latitudes
(Canada, Alaska, Norway) and for JSBACHv3 too large along the Eurasian steppe belt. For the latter region this may be related
to the slightly longer growth period in JSBACHv3 because of the earlier rise of northern hemisphere spring temperatures (see
Fig. 4 bottom). The wide spread and very similar significance pattern between the JSBACH versions reveals that JSBACH is
systematically biased independent of the atmosphere model.

Overall, the bias is small in regions that are only sparsely covered with vegetation (e.g. northern Africa, Middle East, north-
eastern Asia and central North-America) and in regions with strong bias the LAI is generally too large, as is also seen in the
scatter plots of Fig. A5, where the center data cloud is at an LAI bias of about 1. Clearly visible, particularly for JSBACHv3,
is a positive LAI bias in tropical and subtropical regions with high relative soil water levels (compare Fig. 10), which is insofar
plausible as only for good growing conditions the LAI may overshoot. In other regions a spatial correlation between LAI bias
and relative soil water level is not as clear. The general higher RSM in JSBACHv3 as compared to JSBACHv4 is obviously a
consequence of its higher precipitation (Fig. 7).

For a more detailed analysis of the bias patterns one must note that JSBACH uses different parameterizations for its vegeta-
tion types to calculate the seasonal and multi-annual dynamics of the LAI, i.e. to describe their phenology. These parameter-
izations differ in particular by their dependence on growth conditions (temperature, relative soil water and NPP). To analyze
the origin of the biases it is therefore necessary to consider each region with its prevailing vegetation types separately. Such
analysis (not shown) reveals only weak correlations between growth conditions and the diagnosed LAI biases (except for the
above named RSM). It seems unlikely that this is caused by several environmental conditions affecting the LAI dynamics si-
multaneously in a way that correlations cancel. More likely the LAI biases are caused by shortcomings in the parameterizations
of the phenology, for example, by an unlucky choice for the combination of the many parameter values like spring growth rate
for LAI and maximum length of growth season, or by more structural deficiencies like the missing coupling between LAI and
leaf biomass in this type of phenology model.

However, for the LAI bias of JSBACHv4 in Australia we can indeed identify environmental causes. Australia's canopy is es-
pecially in the areas with too high LAI dominated by shrubs and C4 pasture whose phenology is calculated by the parametriza-
tions 'raingreen phenology' and 'grass phenology', respectively. The LAI of both phenology parametrizations depends on the



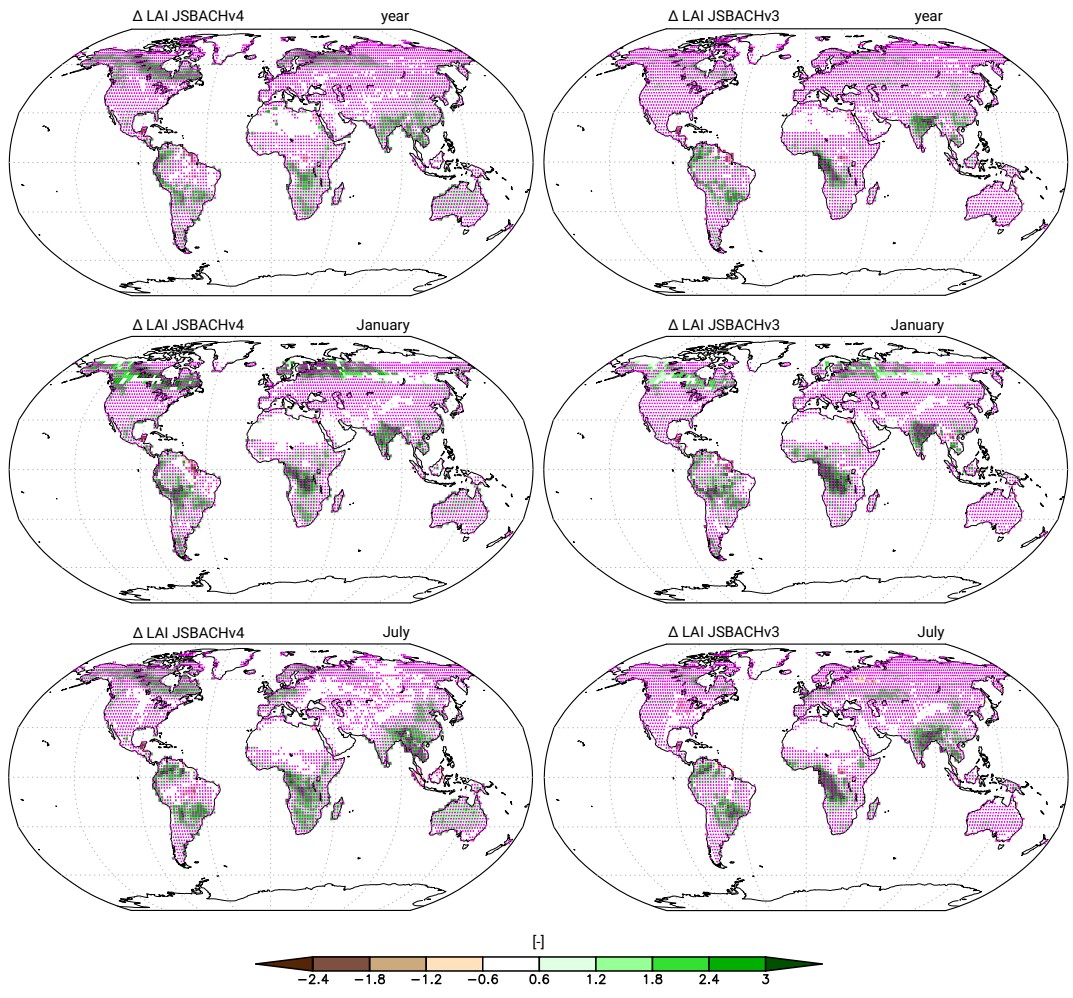

**Figure 9.** Bias in Leaf Area Index (LAI) compared to MODIS data for JSBACHv4 (left) and JSBACHv3 (right) averaged over the years 2001 to 2014. Shown are the annually averaged bias and those for January and July. Pink dots indicate grid boxes where the bias is significant (p=0.05) according to an independent two-sample t-test.

relative soil water level and NPP of the model vegetation. The grass phenology additionally depends on air temperature of the
lowest atmospheric level. As the surface temperature in this area is too low throughout the year (Fig. 5) this is most likely not
the cause for the positive LAI bias. In terms of simulated relative soil moisture (Fig. 10) Australia is not particularly dry. There
is sufficient soil water available for plant growth in the annual average as well as for January and July (values range between
0.4 and 0.5). Particularly in July the dry season of central and northern Australia is not at all represented in the simulations.
As the model bias in NPP is rather weak for July (see Fig. 12) the overly wet soils should be the main reason for the too high
LAI. For January, however, in reality the soil water does not limit plant growth as in July. Therefore, even though our models



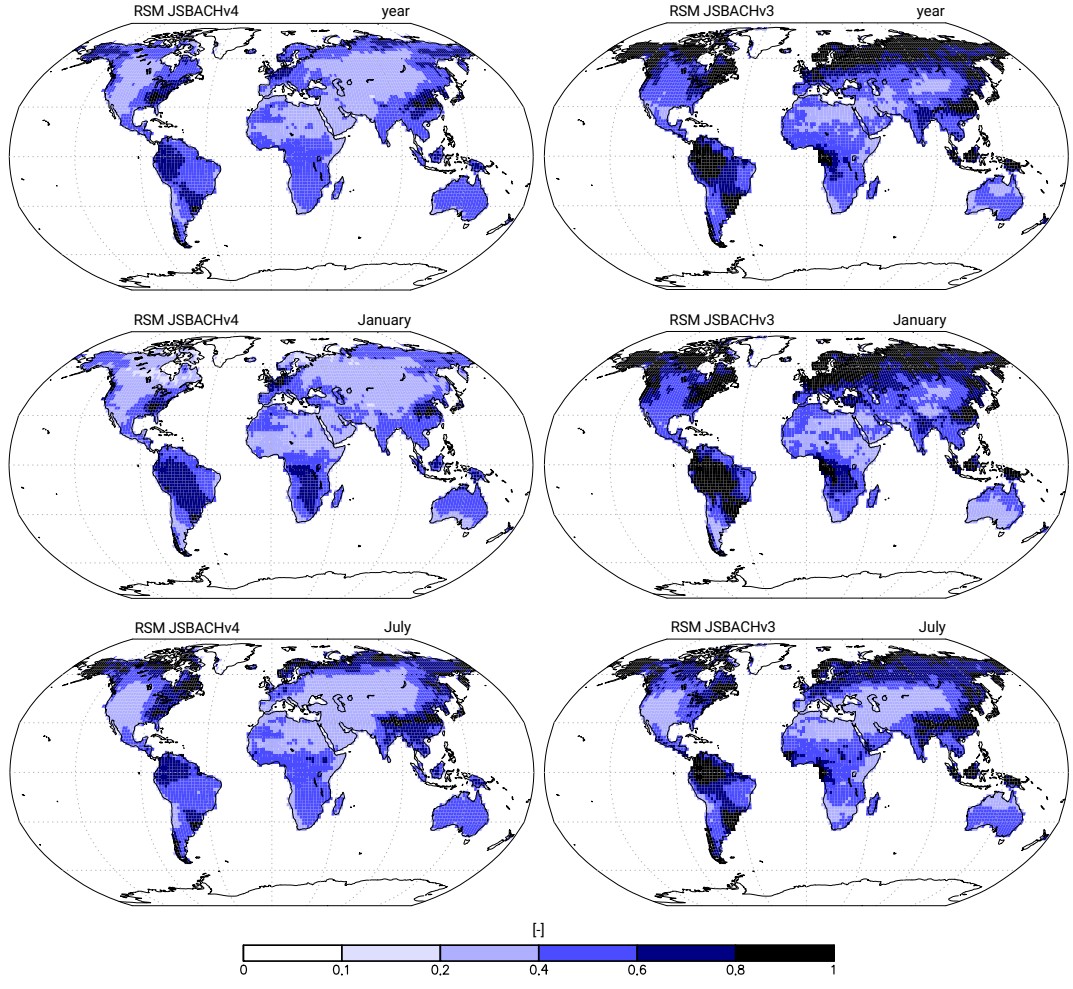

**Figure 10.** Relative Soil Moisture (RSM) in JSBACHv4 (left) and JSBACHv3 (right) averaged over the years 2001 to 2014. RSM is calculated as the ratio between total amount of soil water above wilting point relative to the maximum possible amount of soil water above wilting point.

exhibits too much soil moisture in January, this might not cause the too high LAI. Hence, in this case the positive NPP bias could be the cause for the LAI bias and not vice versa.

In summary we find that: (1) The LAI bias pattern are strongly correlated between JSBACHv4 and JSBACHv3. (2) The LAI is in general high biased which occurs in regions with strong vegetation cover. (3) In Australia for JSBACHv4 the main reasons for the high biases might be the overly wet soils in July and the too high NPP in January. (4) Except for Australia, the LAI biases are probably caused by shortcomings in the parameterizations of the phenology themself and not by the growth conditions.





**Table 5.** Spatial Spearman rank correlations (rho) between the JSBACHv4 and JSBACHv3 biases of LAI against those of FAPAR. In brackets the corresponding Pearson coefficients.

| | JSBACHv4 | | JSBACHv3 | |
|---|---|---|---|---|
| | LAI vs. FAPAR (Kazakhstan) | LAI vs. FAPAR | LAI vs. FAPAR (Kazakhstan) | LAI vs. FAPAR |
| Year | 0.784 (0.766) | 0.640 (0.683) | 0.831 (0.835) | 0.784 (0.798) |
| January | 0.324 (0.285) | 0.505 (0.579) | 0.413 (0.329) | 0.780 (0.818) |
| July | 0.925 (0.939) | 0.732 (0.725) | 0.879 (0.888) | 0.539 (0.632) |

Kazakhstan refers here to the square between longitude 46° to 87° and latitude 40° to 54°N.

## 3.5 Fraction of Absorbed Photosynthetic Active Radiation (FAPAR)

In general FAPAR biases are strongly correlated with the LAI biases (see Tab 5). For JSBACHv3 the FAPAR bias pattern
follows in the tropics and in regions of austral and boreal summer largely the bias pattern of LAI (compare Figs. 9 and 11),
as can be expected from FAPAR being typically larger for deeper canopies. For JSBACHv4 this correlation is weaker but still
strong in the yearly average (see Tab 5). Other possible source of FAPAR deviations are clouds because they might alter the
incoming PAR or the fraction between its direct and diffuse parts. However, the comparison with the bias in Total Cloud Cover
(TCC) (see Fig. 6 and Table 4 ) reveals no general hints for that. FAPAR deviations could also be caused by the VIS soil albedo
which is used in the JSBACH canopy radiation model as lower boundary condition. (The biases in the JSBACH soil albedo
maps were already discussed above.)

   The deviations in FAPAR of the two JSBACH versions (Fig. 11) clearly differ from each other, as also reflected in the
medium correlation values around 0.4 (see Fig. A6). The main differences occur over Kazakhstan, India, south China and
central Africa. The difference over Kazakhstan (around 50°N) is obviously caused by the above described precipitation-RSM
feedback in July. Differences in India, south China and central Africa occur because the high-biased LAI in JSBACHv4 does
not cause the same FAPAR high-bias in these regions as in JSBACHv3. As the VIS soil albedo is the same in both model
versions this can not be the explanation. Unfortunately our simulation outputs do not provide the necessary information about
incoming PAR to verify an atmospheric origin.

## 3.6 Net Primary Production (NPP)

Except for the northern mid latitudes in northern hemisphere summer, the patterns in NPP biases are very similar between the
two JSBACH versions (Fig. 12 and Fig. A7). This is also demonstrated by the rather high correlations in Table 3 for Januaries
(∼0.8) and throughout the year (∼0.75) while there are lower correlation values for the Julys (∼0.5). In general NPP shows
strong correlations with Relative Soil Moisture (RSM) (Table 6). Only for JSBACHv3 in January they are nearly uncorrelated
which is due to the fact that north of 60° there is no NPP at that time but JSBACHv3 has a lot of RSM. However, in general the
bias correlation between NPP and LAI is much weaker.





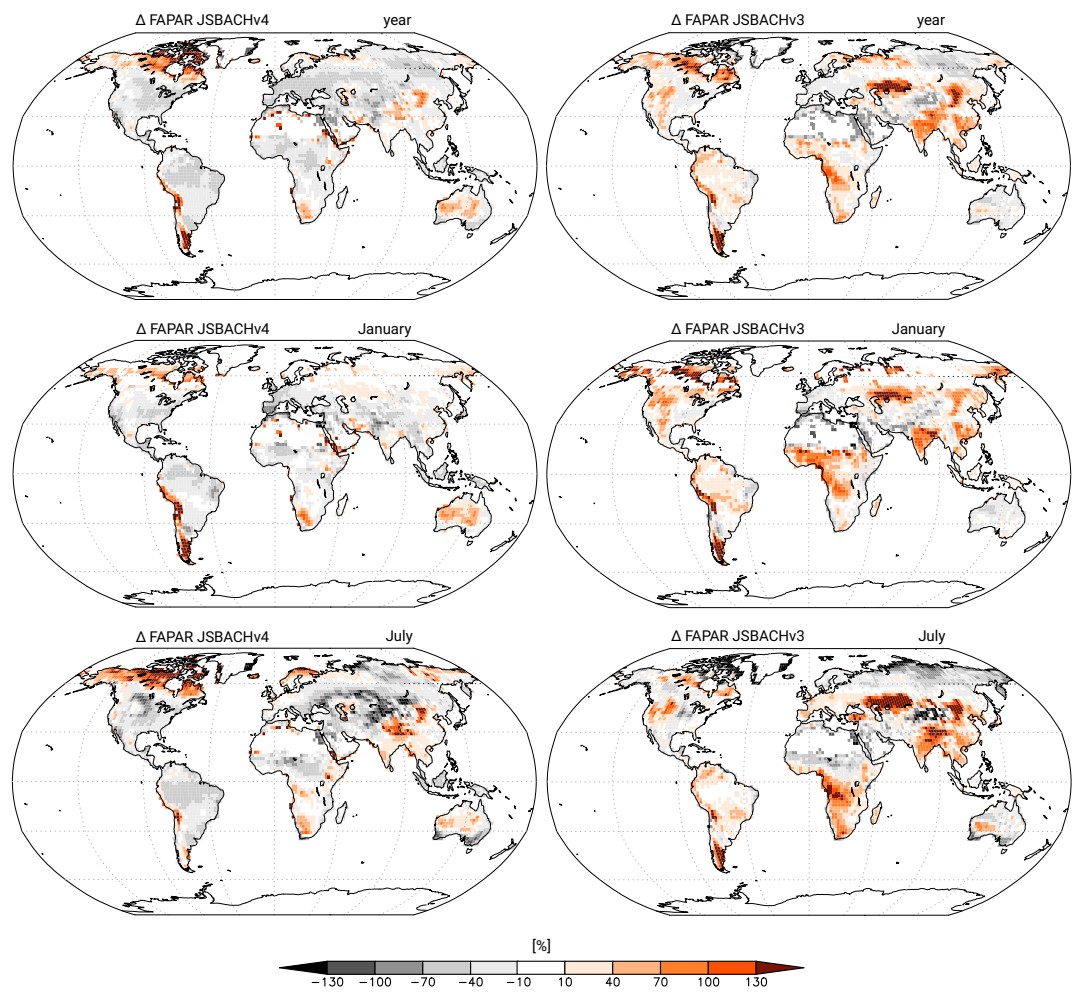

**Figure 11.** Bias in the Fraction of Absorbed Photosynthetic Active Radiation (FAPAR) of JSBACHv4 (left) and JSBACHv3 (right) compared to MODIS FAPAR. Shown are the annual, January and July means over the years 2001 to 2014 in percent to MODIS FAPAR.

**Table 6.** Spatial Spearman rank correlatations (rho) between the JSBACHv4 and JSBACHv3 biases of NPP against those of LAI, FAPAR and RSM.

| | JSBACHv4 | | | JSBACHv3 | | |
|---|---|---|---|---|---|---|
| | NPP vs. LAI | NPP vs. FAPAR | NPP vs. RSM | NPP vs. LAI | NPP vs. FAPAR | NPP vs. RSM |
| Year | 0.197 | 0.270 | 0.823 | 0.397 | 0.550 | 0.739 |
| January | 0.279 | 0.348 | 0.573 | 0.506 | 0.553 | 0.026 |
| July | 0.400 | 0.362 | 0.789 | 0.435 | 0.366 | 0.793 |



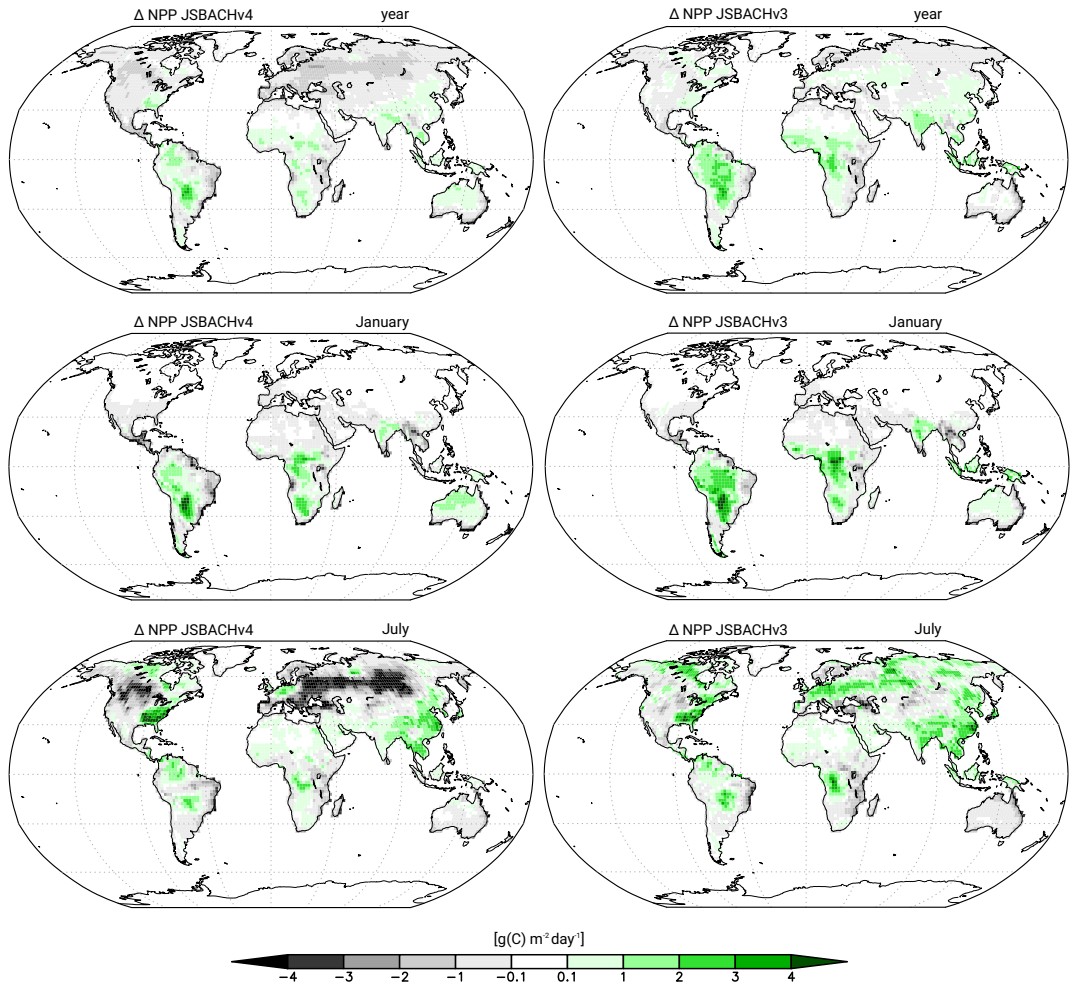

**Figure 12.** Bias in Net Primary Production (NPP) for JSBACHv4 (left) and JSBACHv3 (right) compared to MODIS NPP. Shown are the annual, January and July means over the years 2001 to 2014. Note that significance is not shown because except for the Sahara region all biases are worldwide significant (p=0.05) according to an independent two-sample t-test.

The above named opposing northern mid latitudes NPP biases between the JSBACH versions in July have several reasons. For JSBACHv4 it is obvious that too low precipitation and FAPAR lead to the strong negative NPP bias (compare with Fig. 7 and Fig. 11). This bias is a side effect of the precipitation-RSM feedback already explained above (similar to the LST bias for this region and time). For JSBACHv3 the bias is caused by a too high LAI leading to a too high FAPAR (compare with Fig. 9 470 and Fig. 11).

That the NPP bias is vanishing in winter for regions with seasonal vegetation is trivial, so that only the patterns in summer need explanation. For both JSBACHv4 and JSBACHv3 in the tropics and in the regions of boreal and austral summer the bias pattern in NPP largely follows that of LAI (compare Figs. 9 and 12), which is plausible in view of the already diagnosed





positive LAI bias. The correlations between the NPP and LAI biases are not high (Table 6) as there are no correlations for
the respective winter part of the globe (compare Figs. 9 and 12). The same accounts for the correlations between the NPP and
FAPAR biases although primary production directly depends on FAPAR.

In summary we find that: (1) The NPP bias pattern highly agree between JSBACHv4 and JSBACHv3 and the strongest
mismatch is caused by the July precipitation-RSM feedback of JSBACHv4. (2) The NPP deviations are strongly correlated
with RSM.

**3.7 Water-Use-Efficiency (WUE)**

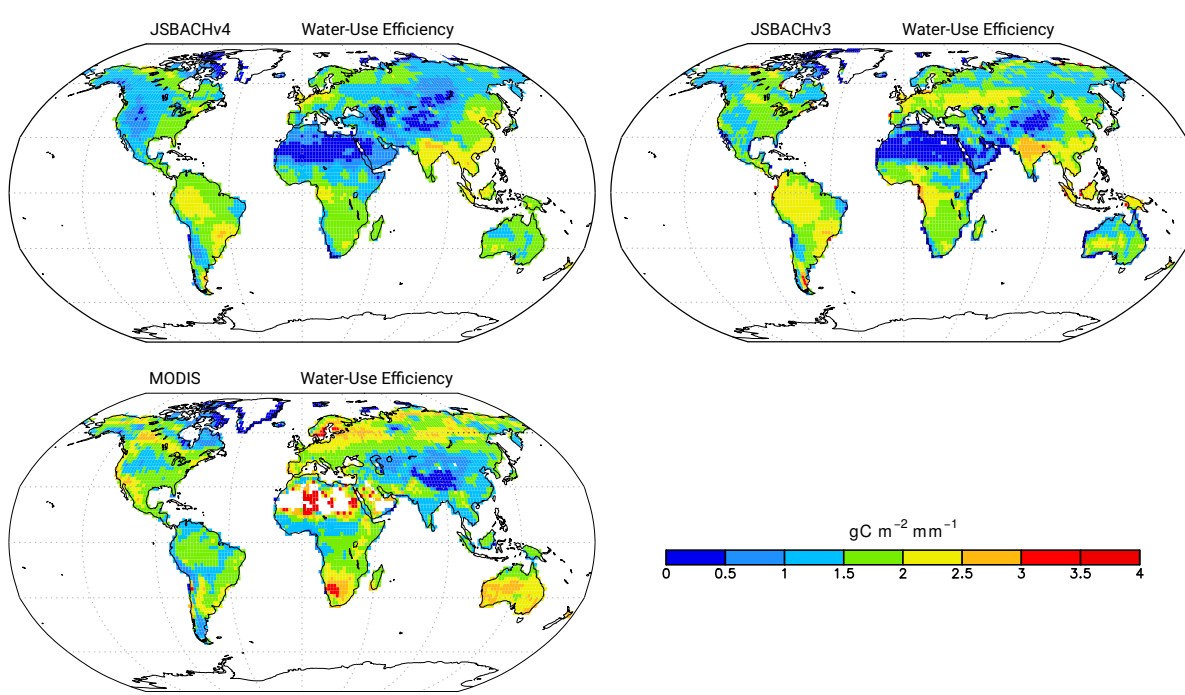

**Figure 13.** Water-Use-Efficiency (WUE). JSBACHv4 (top left), JSBACHv3 (top right) and MODIS (bottom left). All values are averaged
over the years 2001 to 2014. The color scale is chosen to match the one used in Figs. 1 and 2 of the study by Sun et al. (2016).

To discuss WUE as simulated by JSBACHv3 and JSBACHv4, we show simulation results in two ways: as geographic
distribution (Fig. 13), and in climate space (Fig. 14). In both plots colors are chosen to be consistent with the color coding of
the respective Figs. 1 and 2 of Sun et al. (2016) showing WUE obtained from observation data and also from simulations with
other land models.

Looking first at the geographical distribution (Fig. 13), WUE turns out to be worldwide slightly weaker in JSBACHv4 than
in JSBACHv3, but otherwise the pattern is similar. A major difference between the model versions is the rather low WUE of



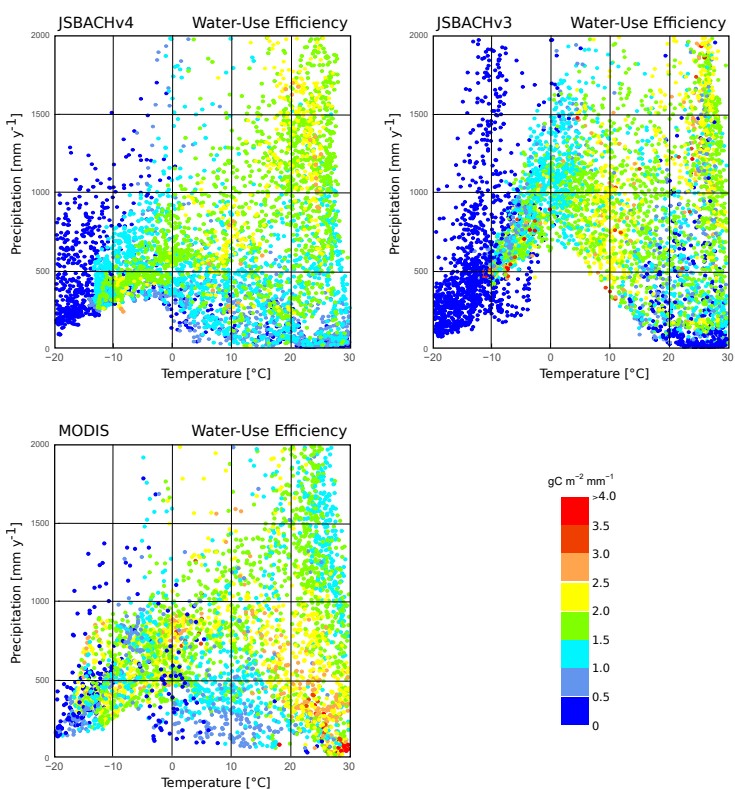

**Figure 14.** Water-Use-Efficiency (WUE). JSBACHv4 (top left), JSBACHv3 (top right) and MODIS (bottom left) for each grid cell plotted against annual 2m-temperature and precipitation. All values are averaged over the years 2001 to 2014. Color scale as in Fig. 13.

JSBACHv4 found in the agricultural belt along the Eurasian steppe regions, where in the models vegetation consists of crops. (Note, there is no correlation with the additional C4 crops PFT of JSBACHv3.) Causal for this is the lower GPP of JSBACHv4 in this region (not shown, but also visible somewhat less pronounced in NPP Fig. 12).

The large scale pattern is best characterized by noting that JSBACH reveals low values for WUE in western North America, north-eastern Canada and in the broad belt ranging from the Sahara deep into inner Eurasia; elsewhere WUE is considerably larger. The reason for this pattern is obvious from Fig. 10, showing the relative soil water content above wilting point. RSM is therefore a measure of the water stress experienced by the vegetation in JSBACH: regions of high water stress broadly match the regions of low WUE. Accordingly, it is the low plant productivity that determines the regions of low WUE. Nevertheless,

in the hot deserts – that make up large parts of the regions of low WUE – high potential evaporations surely adds to WUE being small there.

This large scale pattern is also broadly consistent with the MODIS WUE (Fig. 13 bottom left). In contrast to JSBACH MODIS exhibits the highest global values over the dry areas in northern and southern Africa and in Australia. Sun et al. (2016)





calculated the WUE from MODIS GPP and JUNG GPP (their Figs. 1a and 1b, the invoked evapotranspiration data are the same). The WUE calculated from their MODIS dataset is unsurprisingly very similar to our MODIS WUE and shows the same high values in these dry areas. However, the JUNG data (Sun et al. (2016) Figs. 1b) show rather low values for these regions and are therefore in much better agreement with JSBACH. A general disagreement between all observation based datasets and JSBACH occurs in India: The too high WUE simulated for India is caused by the PFT 'C3 crops': In this region C3 types ('C3 grass', 'C3 pasture' and 'C3 crops') are the most productive PFTs and among these 'C3 crops' are in terms of area the most dominant PFT.

Regarding the distribution of our WUE in 'climate space' spanned by temperature and precipitation (Fig. 14) the results of JSBACHv3 are very similar to JSBACHv4. ECHAM6 (JSBACHv3) shows much higher precipitation in the temperature range around -15°C and 15°C. As a consequence JSBACHv4 fits much better in this range than JSBACHv3 when compared to our MODIS data or to Sun et al. (2016) Fig. 2 (a) and (b). The above mentioned much higher global values over dry areas in MODIS as compared to JSBACH are also visible here. MODIS has its highest values in the high temperature low precipitation corner which is the opposite in both JSBACH versions. The JSBACH Indian high bias is visible at 25°C and precipitation above 1000 mm/y.

## 4  Discussion and conclusions

### Albedo

As compared to the JSBACHv3 results JSBACHv4 shows very similar biases (and significances). Overall the representation of the albedo in JSBACHv4 is as good as for JSBACHv3.

Nearly all albedo deviations of JSBACH are significant. In Antarctica and Greenland the albedo deviations are uniformly too high for NIR and uniformly too low for VIS. Therefore, these deviations can be mitigated without side effects within the JSBACH calculations of the glacier albedo. For glaciers albedo values are used that vary as function of temperature continuously between fixed minimum and maximum values. Adapting these minimum and maximum values alone should already reduce the bias and tuning the temperature dependency could further improve the results. In the northern mid latitudes the January albedo of JSBACH is low biased and is largely caused by a too small snow cover. Most probably the already existing initial warm bias of ICON-A in these areas (see LST) is enhanced by the snow-albedo feedback. Tuning the snow scheme in JSBACH towards more snow might compensate the warm bias of ICON-A. However, there could be undesired side effects, such as a cold bias in eastern North-America, a delay of the spring point or too little soil water.

The JSBACH albedo bias in other regions are at least partly caused by canopy or soil albedo biases. Another source could be that the scheme merging them to the overall albedo causes additional deviations. For example, wrong fractions of bare ground below vegetation that is "visible" for the sun or wrong fraction of soil below the vegetation which is covered by stems but not by the LAI. However, we found that the JSBACH albedo bias in these regions can be largely explained by the canopy and soil albedo of the model. In the present model configuration the values for canopy are calculated from PFT specific leaf albedo values, which themselves do not account for seasonality. The NIR and VIS soil albedos in the present model configuration





are taken from prescribed maps, which also do not account for seasonality. The albedo values for the leaves and for the soil are derived by linear regression of the FAPAR on total surface albedo (described in Otto et al. (2011)). Thus, both the canopy and soil albedo values within JSBACH are based on the 2001 to 2004 average of the MODIS MOD43C1 CMG Collection 4

Albedo Product. We showed that the differences between this product and the January/July values of the MODIS product used for our assessment already explain large parts of the JSBACHv4 albedo deviations. Therefore, as a first mitigation step the MODIS database for JSBACH should be updated and a seasonality for leaves and soil should be introduced in the JSBACH albedo scheme. Only after that it would make sense to investigate improvements of the scheme that merges snow, canopy and soil albedo to the overall albedo.


**Land Surface Temperatures**

Overall, JSBACHv3 and JSBACHv4 show very similar temperature biases. The global mean LST simulated by JSBACH match well with observational data, especially for JSBACHv4 (see Fig. 4 top). However, this is only because strong regional biases mutually cancel each other. The overall zonal bias pattern of both models hint to a strong atmospheric bias contribution. We

find for both models a large scale warm bias throughout the year over central Asia which is also significant in the zonal averages. One reason for this is the warming effect of the TCC in January, which is enforced by a too small snow cover and the snow-albedo feedback in this region. For JSBACHv4 an additional feedback between low precipitation and low RSM causes a strong warm bias in July. It is almost certain that this feedback is caused by the combination of the 5-layer snow and soil water freezing schemes implemented in JSBACHv4. The effect of this scheme on the JSBACH results is analysed in detail by

Hagemann et al. (2016) and is in very good agreement with our results. For the predecessor model ECHAM5 similar northern hemisphere winter and summer warm biases as in our results are already shown in Ozturk et al. (2012) while the January and July biases shown in Piani et al. (2010) agree only partly (Kazakhstan and northern China). We also find too high temperatures over the Antarctic and Greenland ice sheets and cold-biased regions in S-America, northern- and southern Africa, Australia, India and China (all significant). These biases are at least partly caused by albedo deviations. As the central Asian, Antarctic

and Greenland warm biases are directly connected to albedo deviations they might be mitigated by the model adaptations proposed above for the albedo.

**Terrestrial Water Storage**

We analysis the agreement of alterations of TWS changes in the course of the year between JSBACHv4 and GRACE. We

found that the agreement is best for regions where precipitation is not very sensitive to the exact representation of atmospheric conditions in the model (i.e. regions where precipitation underlies dominant atmospheric processes like deep convection). An implication of this result is that the representation of TWS changes in the course of the year are less dependent on soil properties like field capacity or porosity than on the atmospheric conditions. Thus, from this point of view the enhancement of soil properties is not a main focus for the development of JSBACHv4.






**Leaf Area Index**

The patterns in LAI biases are very similar between both model versions. In general the LAI is significantly too high in JS-BACH. For JSBACHv4 in Australia the LAI deviations for July is probably caused by too much soil moisture and for January by too high NPP. Therefore, for July, when our model does not represent a dry season in Australia, the LAI bias would be fixed

by improving relative soil moisture. For January the LAI bias might be reduced with a mitigation of the NPP bias. Except for JSBACHv4 in Australia and the general too high RSM the LAI biases and the corresponding growth conditions are correlated only weakly. However, as our growth conditions (temperature, relative soil water and NPP) exhibit strong shortcomings a first step is to enhance them. As a second step one could reduce the LAI by adapting the parameters for the phenology growth and death rates. As the LAI has strong influence on other processes (sensible and latent heatflux, GPP and dark respiration, canopy

albedo) it would be at the same time necessary to adapt these processes to the lower LAI. Nabel et al. (2020) show that the introduction of forest age classes in JSBACHv4 also reduces the LAI bias. However, this comes with additional computational costs for the model.

**Fraction of Absorbed Photosynthetic Active Radiation and Net Primary Production**

We find for both model versions that FAPAR deviation are mainly caused by LAI biases. Overall all NPP deviations in JS-BACHv4 can be associated with water stress problems, FAPAR and LAI. The high-biased NPP would be bettered nearly everywhere with the above named reduction of the LAI. With this reduction also the FAPAR biases would be reduced and contribute to an improved NPP (e.g. in southern Africa and India). Our NPP biases are strongly correlated with RSM respectively with water stress. Higher water stress in S- and eastern N-America and central Africa would further reduce our NPP deviations.

However, to avoid the strong JSBACH4 low-bias in July for the mid latitudes it would be necessary to enhance the ICON-A precipitation in this area which is a consequence of the above named precipitation-RSM feedback.

**Water-Use-Efficiency**

Overall the WUE of our models looks very reasonable. In the Sahel Zone JSBACH WUE seems to be too low as compared to

MODIS data, but it is possible that MODIS is too high here, especially as neither the JUNG data nor the models used in Sun et al. (2016) show the same high bias. In India our WUE is too high. One reason for this is the strong assimilation from C3 crops. This is also visible in the NPP high-bias for India. However, a high-biased productivity of C3 crops can not be found somewhere else. Another reason might be that JSBACH does not incorporate irrigation in our simulations, which is substantial in India and enhances evapotranspiration. As a result WUE in JSBACH should be higher than observed. For JSBACHv3 an irri-

gation scheme was already implemented (de Vrese and Stacke, 2020) and could be transfered to JSBACHv4. JSBACHv4 WUE is low-biased in central Asia which is probably caused by our low-biased FAPAR. WUE in JSBACHv4 is generally somewhat weaker than in JSBACHv3 but the geographical patterns are similar. The overall temperature and precipitation range of ICON-A+JSBACHv4 and ECHAM6+JSBACHv3 is similar to observational data, but especially ECHAM6+JSBACHv3 reveals too high precipitation in the temperature range -15° to 15°C. Therefore, JSBACHv4 performs better than JSBACHv3 when com-

pared to observations. With a general reduction of the LAI as proposed above and the associated reduction of GPP the WUE





of JSBACH might get low biased.

**Perspectives of JSBACHv4**

Our comparison with JSBACHv3 shows that the performance of JSBACHv4 is overall quite similar to JSBACHv3. Differ-
ences can partly be attributed to the different atmospheric conditions generated from and in interaction with the atmospheric
host models ICON-A and ECHAM6, e.g. differences in WUE could be traced back to differences in precipitation. In terms
of process descriptions, JSBACHv3 and JSBACHv4 are almost identical. The only notable difference in simulation results
arising from such structural differences concerns regional differences in surface temperature via modified feedbacks with the
atmosphere that we attribute to the implementation of a multi-layer snow model and freezing soil water in JSBACHv4. A few
options to improve JSBACHv4 have been outlined in this paper. It should be noted that the versions of ICON-A and ICON-
Land/JSBACHv4 used for this study constitute the first milestone in the development of the new ICON Earth System Model
ICON-ESM. The flexibility of the ICON-Land infrastructure provides the basis for further developments and improvements
as the land component of ICON-ESM. The model configuration of the AMIP experiments described in this paper only use the
physical and biogeophysical components of JSBACHv4. The carbon cycle and dynamic vegetation processes of JSBACHv3
have also already been ported to JSBACHv4 and can already be used to model the fully closed carbon cycle within ICON-ESM
together with the ocean biogeochemistry module. The evaluation of these processes will be addressed in future work.

*Code and data availability.* The source code, scripts, and necessary forcing and boundary data of ICON-ESM-V1.0 used in this study
for AMIP simulations with ICON-A is downloadable from https://doi.org/10.35089/WDCC/RUBY-0_ICON-_ESM_V1.0_Model (click
on "Find data"). Note that by downloading the included license agreement (see https://mpimet.mpg.de/en/science/modeling-with-icon/
code-availability) is accepted. The primary data and scripts used to produce the figures are downloadable from https://cera-www.dkrz.
de/WDCC/ui/cerasearch/entry?acronym=DKRZ_LTA_060_ds00008. The data from the MPI-ESM1.2-HR AMIP simulations are available
at the CMIP6 repository of the Earth System Grid Federation (Lorenz et al., 2021) and can be accessed via https://doi.org/10.22033/ESGF/
CMIP6.6463.

**Appendix A**

For direct comparison of the simulation bias of the two model versions we plot in this appendix the difference to observations
of the various assessment variables for each grid cell of one model against that of the other (see e.g. the scatter plot A3). The
statistical values noted in these plots are the intercept and slope of the linear regression, the Pearson correlation $r$, its square
$R^2$ (also called 'coefficient of determination'), and the Spearman rank correlation $rho$. In addition, to obtain a rough idea how
differences in simulation bias varies with region, we colorize the data points according to their broad geographical origin. For
this we distinguish four regions (Fig. A1):

*Polar Zone* (black): 90° to 60° and -90° to -60°

*Temperate Zone* (red): 59,9° to 37.9° and -59,9° to -37.9°





*Subtropical Zone* (yellow): 38° to 15° and -38° to -15°

*Tropical Zone* (green): 14,9° to -14,9°

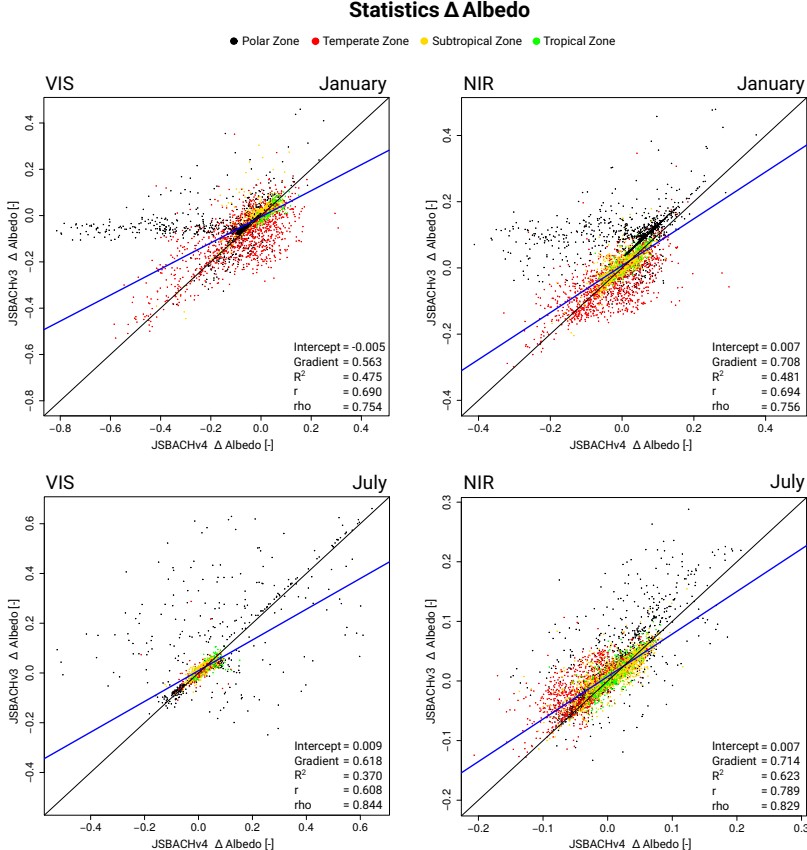

**Figure A1.** Zones of the scatterplots.

**Figure A2.** Comparison of simulated albedo bias (Δ Albedo) between the two JSBACH versions. Plotted is for each grid cell the albedo bias of JSBACHv3 against that of JSBACHv4. The bias data are the same as in Fig. 1. Shown are the averages of Januaries (top), and Julys (bottom) for VIS (left) and NIR (right) albedo. For more details see section 2.4.



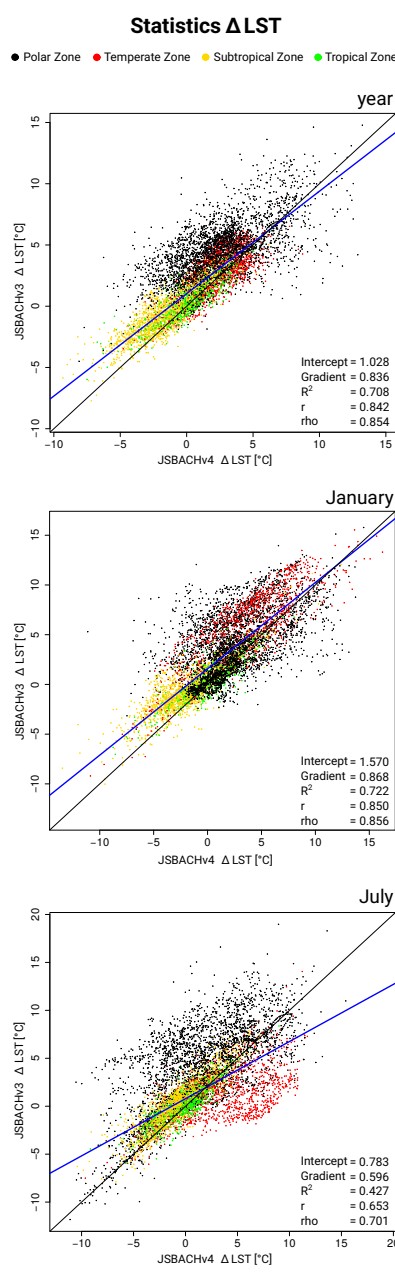

**Figure A3.** Comparison of simulated LST bias ($\Delta LST$) between the two JSBACH versions. Plotted is for each grid cell the LST bias of JSBACHv3 against that of JSBACHv4. The bias data are the same as in Fig. 5. Shown are the annual averages (top), average of Januaries (mid), and Julys (bottom). For more details see section 2.4.



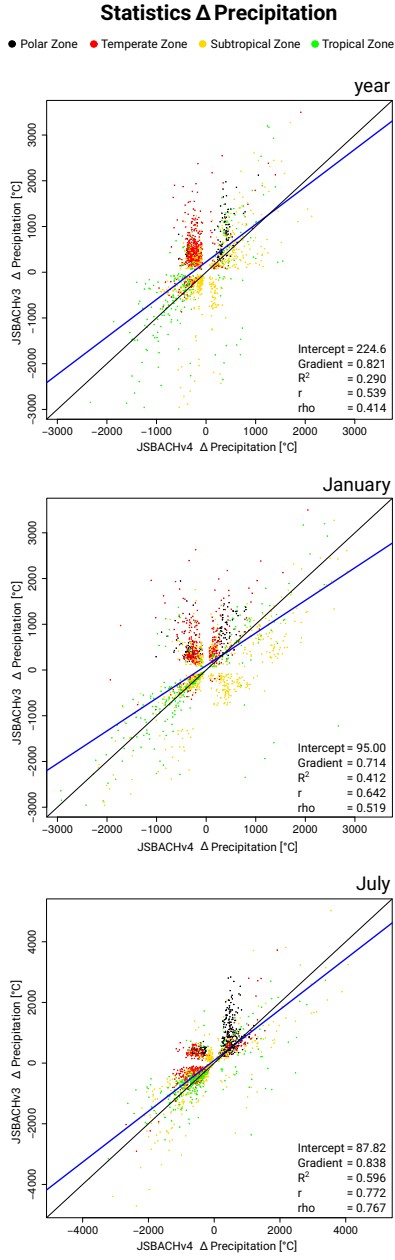

**Figure A4.** Comparison of simulated precipitation bias (Δ Precipitation) between the two JSBACH versions. For each grid cell the precipitation bias of JSBACHv3 is plotted against that of JSBACHv4. The bias data are the same as in Fig. 7. Shown are the annual averages (top), averages of Januaries (mid), and Julys (bottom). For more details see section 2.4.



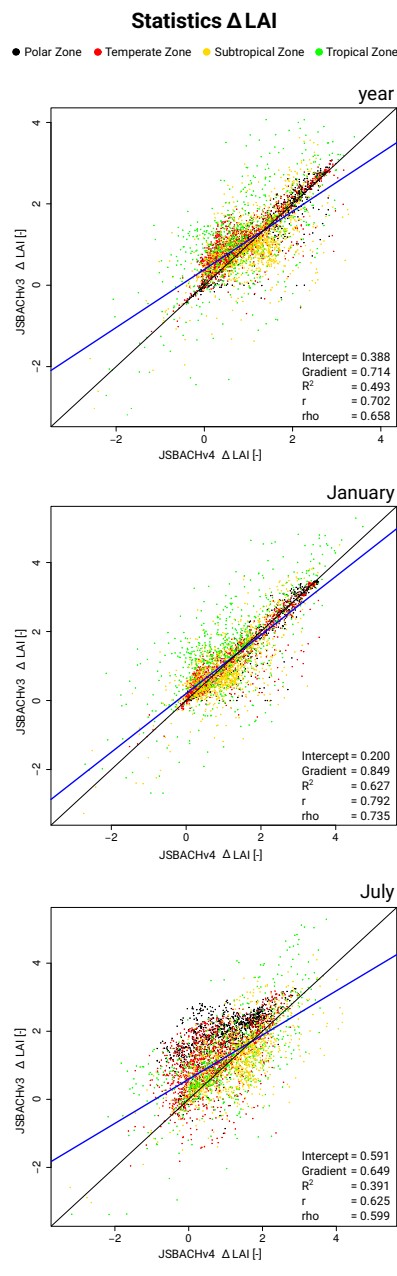

**Figure A5.** Comparison of simulated LAI bias (Δ LAI) between the two JSBACH versions. For each grid cell the LAI bias of JSBACHv3 is plotted against that of JSBACHv4. The bias data are the same as in Fig. 9. Shown are the annual averages (top), averages of Januaries (mid), and Julys (bottom). For more details see section 2.4.



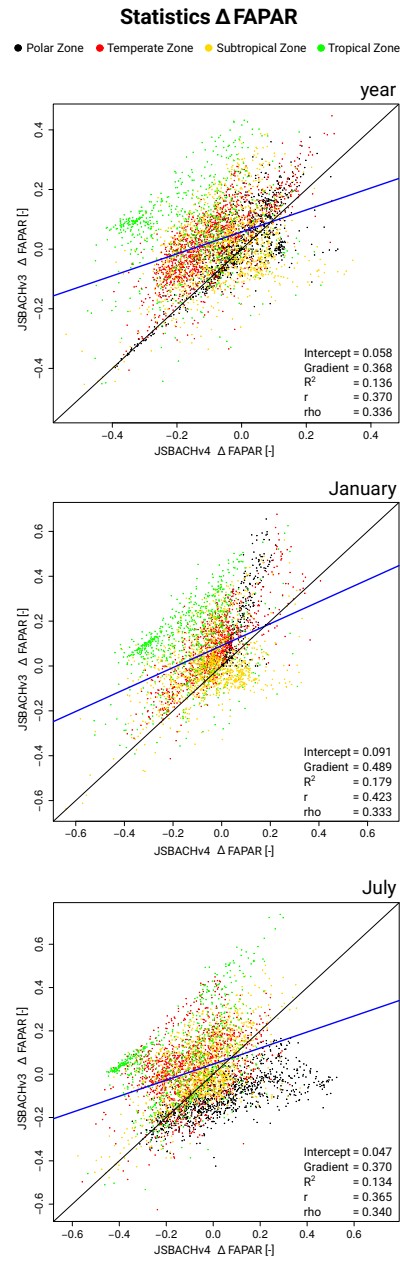

**Figure A6.** Comparison of simulated FAPAR bias (Δ FAPAR) between the two JSBACH versions. For each grid cell the FAPAR bias of
JSBACHv3 is plotted against that of JSBACHv4. The bias data are the same as in Fig. 11. Shown are the annual averages (top), averages of
Januaries (mid), and Julys (bottom). For more details see section 2.4.



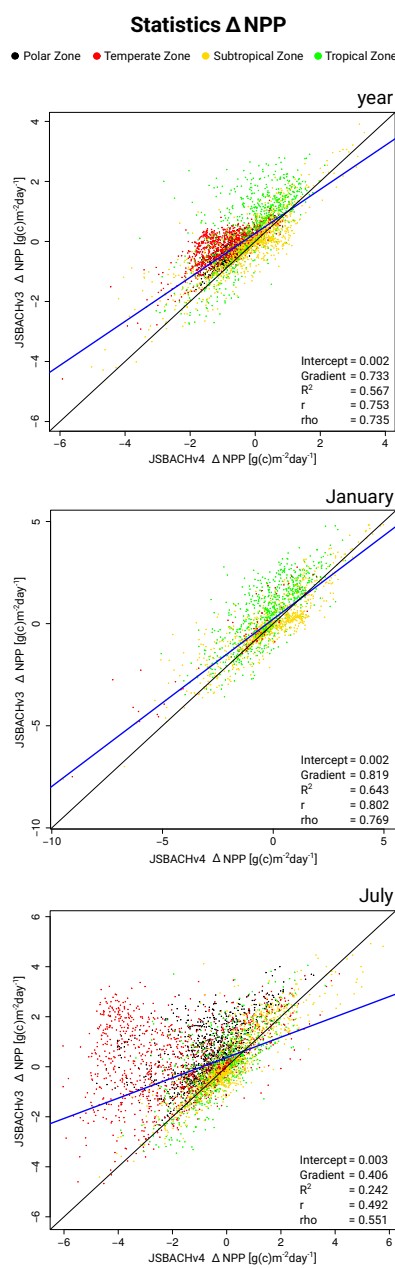

**Figure A7.** Comparison of simulated NPP bias (Δ NPP) between the two JSBACH versions. For each grid cell the NPP bias of JSBACHv3 is plotted against that of JSBACHv4. The bias data are the same as in Fig. 12. Shown are the annual averages (top), averages of Januaries (mid), and Julys (bottom). For more details see section 2.4.

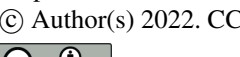



**Table A1.** Spatial Spearman rank correlations (rho) between the JSBACHv4 and JSBACHv3 biases of precipitation against those of LST.

|  | JSBACHv4 | | JSBACHv3 | |
|---|---|---|---|---|
|  | Preci. vs. LST (Eurasia) | Preci. vs. LST | Preci. vs. LST (Eurasia) | Preci. vs. LST |
| Year | -0.416 | -0.543 | 0.283 | 0.262 |
| January | -0.090 | -0.403 | 0.278 | 0.312 |
| July | -0.638 | -0.413 | -0.169 | -0.031 |

Eurasia refers here to the square between longitude 0° to 184° and latitude 35° to 90°.



*Author contributions.* R. Schneck performed the simulations, and prepared the data and plots. All authors contributed to the design, data interpretation and text of the study.

*Competing interests.* There are no competing interests.

*Acknowledgements.* We thank Dr. Stefan Kern for his advice concerning the handling of satellite remote sensing data and the German Climate Computation Center (DKRZ) for providing IT resources and support. Furthermore, we like to thank Dr. Jürgen Bader for his internal
review of our manuscript and his constructive comments.





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
