# Peer review of "Assessment of JSBACHv4.30 as land component of ICON-ESM-V1 in comparison to its predecessor JSBACHv3.2 of MPI-ESM1.2"

_Geoscientific Model Development, 2022_

## Author Comment (AC1)

**Response to Referees**

We thank both reviewers for reading our manuscript and providing constructive criticism. We are pleased with the recommendations for publication from the reviewers: referee#1 *"In general, the paper is written and structured well and fits into the scope of the journal."* and referee#2 *"I recommend the publication of the manuscript after minor revisions."*

In the following we address each of the comments of the reviewers point-by-point. For that we paste the original comments in red and our response in black. Our corresponding changes to the manuscript are presented cursive and as separate blocks.

**Referee#1**

The authors evaluate the performance of the new JSBACHv4 biosphere model in comparison with the JSBACHv3 model in the context of the new ESM model developed at MPI Hamburg. In this paper they have a focus on fast (sec to year timescale) processes. In general, the paper is written and structured well and fits into the scope of the journal. Because of the focus on evaluation than on model description the paper lacks mathematical description of the model changes. Perhaps adding some formulae would be beneficial. But I see the constraints due to the length of the paper.

Indeed our focus is on assessment and not on model description as the scientific contents of the new JSBACHv4 are a subset of JSBACHv3 and that model is comprehensively documented (Reick et al, 2021) with all mathematical detail. We emphasize this better in the revised version by adding "(documented in Reick et al., 2021)" in the following sentence (line 100):

*"However, it applies the same parametrizations as JSBACHv3 (documented in Reick et al., 2021) and includes the additional feature of frozen soil water and a five-layer snow scheme (Ekici et al., 2014; de Vrese et al., 2021)."*

*C.H. Reick et al. (2021), JSBACH 3 - The land component of the MPI Earth System Model, documentation of version 3.2, Reports on Earth System Science, 240, https://doi.org/10.17617/2.3279802.

Specific comments:

The model setup has to be described in more detail. The model setup is distributed at different parts in the manuscript (e.g. prescribed ocean SST, prescribed PFT distributions).

Thank you for noting this. Indeed, it is not very fortunate that the description of the model setup spreads over different sections. We reordered the former sections 2.1 and 2.2 and added a new section "Simulation setup" to the methods. Accordingly we had to adapt the former text. As the new section (2.2. Simulation setup) encompasses a considerable amount of text, we refrain from quoting it here and suggest to read it in the newly submitted manuscript.

NPP has been selected for comparison with observational data. It would be interesting to see above-ground biomass as an additional validation data set.

The main distinction between the carbon pools representing plant organs in JSBACH is by turnover time, namely fast and slow compartments. Accordingly, fine roots and leaves are combined in a single carbon pool, and stem and coarse roots also in a common carbon pool. Therefore in the carbon pool model above and below ground carbon are not distinguished and thus, although interesting, the suggested comparison is not feasible for our model.

Due to the prescribed vegetation distribution cover cannot be used as an evaluation. In general is would be interesting to the see the performance of the ESM for a dynamic vegetation without prescribed PFTs. But this would be a topic for a separate publication.

We agree.

The formatting of tables could be improved. The vertical lines in the headers do not correspond. In particular a separator for the two JSBACH versions in the header is missing.

We agree and re-formatted all tables accordingly.

To quantify the bias between simulation and observational data the normalize mean error (NEM) metric might additionally be used (Kelly et al, 2013).

A good suggestion. We calculated the Normalized Mean Error for all assessment variables of the two models (JSBACHv4 and JSBACHv3) and show them in two tables:

**Table A1.** Normalized Mean Error (NME) of JSBACHv4 relative to observations.

|         | Albedo VIS | Albedo NIR | LST   | LAI   | FAPAR | NPP   | WUE   |
|---------|------------|------------|-------|-------|-------|-------|-------|
| Year    | -          | -          | 0.138 | 1.294 | 0.796 | 0.783 | 1.341 |
| January | 0.267      | 0.410      | 0.149 | 1.266 | 0.604 | 0.569 | -     |
| July    | 0.411      | 0.508      | 0.193 | 1.102 | 0.757 | 0.777 | -     |

**Table A2.** Normalized Mean Error (NME) of JSBACHv3 relative to observations.

|         | Albedo VIS | Albedo NIR | LST   | LAI   | FAPAR | NPP   | WUE   |
|---------|------------|------------|-------|-------|-------|-------|-------|
| Year    | -          | -          | 0.147 | 1.430 | 0.669 | 0.658 | 1.406 |
| January | 0.232      | 0.415      | 0.164 | 1.304 | 0.687 | 0.576 | -     |
| July    | 0.400      | 0.452      | 0.176 | 1.310 | 0.688 | 0.653 | -     |

However, as proposed by referee#2 we also added a Taylor Diagram that visualizes the model performance even better. Therefore we added the tables to the appendix and only mention its results in the new "General Performance" section (see answer to the referee#2).

**Referee#2**

The authors evaluate the performance of JSBACHv4 within the ICON ESM and JSBACHv3 within the MPI ESM versus a set of observational data. The authors rely on coupled land-surface-atmosphere simulations with prescribed sea surface temperature and sea ice. Variables evaluated include albedo, Land Surface Temperature (LST), Terrestrial Water Storage (TWS), Leaf Area Index (LAI), Fraction of Absorbed Photosynthetic Active Radiation (FAPAR), Net Primary Production (NPP), and Water-Use-Efficiency (WUE). Biases between model results and observations are substantial in many variables. JSBACHv4 performs similarly to JSBACHv3 (line 604) as the process description are almost identical for both models. This raises the question of why JSBACHv4 was not improved relative to JSBACHv3 to avoid some of the major model biases.

The transition from MPI-ESM to ICON-ESM is an ongoing major effort at our home institution MPI for Meteorology with a history of more than one decade of technical and scientific development. So far the development of ICON-ESM concentrated separately on the major Earth system components (atmosphere, ocean, sea ice, land). Even if the components work well when run in isolation, the combined model typically performs less well because of the unforeseeable effects of the additional degrees of freedom arising from the interactions between the components. This makes it necessary to further tune the combined model, and this is another major effort. Tuning concentrated first on the stability of ocean circulation and the reproduction of historical mean climate. The resulting ICON-ESM Version 1.0 is a major milestone of our model development documented in Jungclaus et al. (2022). Next tuning steps for various other aspects must follow. Even a well equipped institute like ours cannot maintain in parallel two ESMs for years. Therefore research using our new ICON-ESM scientifically started early, well knowing that its development is not finished, but also an unfinished model can be used for serious science when being aware of its particular deficiencies. This is one major reason why the assessment layed down in the submitted manuscript is so important for the many (of in particular the German scientific community) working on land issues with this model. – We hope that this explains why we want to publish this assessment already now and don't wait until biases, that we partly got aware of only because of this assessment, are remedied.

Land carbon cycle and climate-induced biogeographical changes in landcover are not assessed. Modules of JSBACH representing the latter two are switched off. It is a weakness of this study that biogeographical changes and the land carbon cycle modules are switched off and not evaluated as the distribution of plants has an impact on albedo, WUE, etc.

We agree. Nevertheless, we want to point out that for such a major new development as ICON-ESM (see above) the long time scales of biogeographical changes and land carbon turnover stand at the end of the tuning efforts, because first the mean climate must be correct. But unfortunately we are not at this tuning state yet.

I recommend the publication of the manuscript after minor revisions.

We are happy to read this!

Specific comments

1. The authors link biases in albedo and related variables to the applied soil albedo and canopy map (line 326) and fixed minimum and maximum albedo values (line 322) all already used in JSBACHv3. Surprisingly, the authors do not update these features or at least the albedo map in JSBACHv4. This downgrades the manuscript somewhat to a progress report. It seems a necessary next step is to update the albedo module to reduce the large biases in model outcome versus observations. It would make the paper more interesting and useful if these updates in JSBACHv4 would be implemented (maybe this is for computational and personal reasons not possible?). Otherwise, the risk is that this manuscript is outdated very quickly.

We agree but must once more point to the circumstances of model development at MPI: That it will be necessary to update the albedo maps is one of the outcomes of this assessment. We got aware of this only after the many simulations by which the performance of ICON-ESM Version 1.0 is documented in Jungclaus et al. (2022). Moreover, it is this model version that is already scientifically used in our community so that it is important to document these biases.

2. A graphic that summarizes the outcomes of the evaluation concisely is missing. For example, a so-called Taylor diagram should be added to show data-model agreement across variables.

We added a new section "General Performance" to the results where the corresponding Taylor Diagram (see figure below) is shown.

[Figure]

**Figure 1.** Taylor Diagram (Taylor, 2001) of normalized pattern statistics for annual means of our main evaluation variables LST, LAI, FAPAR, NPP and WUE. The diagram contains no values for albedo because during polar winters observations are missing. JSBACHv3 is shown in small dots and JSBACHv4 in big darker dots. Arrows indicate the change from JSBACHv3 to JSBACHv4. The centered pattern root-mean-square error (RMSE) and standard deviations have been normalized by the observed standard deviation of each field before plotting. Correlations are depicted by lines from the origin representing angles relative to the horizontal base line. Standard deviations are represented as arcs around the origin. Normalized centered pattern RMSE values are seen as circular arcs around the value 1 of the baseline (normalization on standard deviation 1.0).

We added the following text (line 355):

*"Taylor (2001) proposed a graphical way to depict the overall performance of a model. The corresponding Taylor Diagram for JSBACHv4 and JSBACHv3 is shown in Fig.1. The LST achieved the by far best agreement of the statistical metrics with observations. These metrics remained nearly unchanged from JSBACHv3 to JSBACHv4. Except for the LST all standard deviations are reduced in JSBACHv4 (especially FAPAR). Overall some statistical metrics of JSBACHv4 improved while others worsened as compared to JSBACHv3. Thus the overall performance remained more or less the same. This is also visible in the NME in Table A1 and A2, where the NME magnitude is very similar between the assessment variables of both models."*

3. A display documenting the seasonal evolution of snow cover is missing. Showing only January snow cover is not enough for a proper evaluation.

We show snow cover only as an auxiliary variable to explain the albedo bias, assessment of snow cover itself is not intended. Indeed we could extend our assessment to snow cover, but we decided to concentrate on variables that picture integrated behaviour across many processes, and for albedo snow cover is only one component. Accordingly, we prefer not to add a separate assessment of snow cover. Obviously we did not make this distinction between main assessment variables and auxiliary variables sufficiently clear in the text and add clarifying remarks (line 144):

*"To shed some more light on the origin of biases in the selected assessment variables, we consider also some auxiliary variables, these are introduced below jointly with the respective assessment variable."*

In addition we updated table 2 and plotted the assessment variables in contrast to the auxiliary variables bold. In each case we think that the interested reader gets at least a first idea of the seasonal course of biases from the two plots for January and July that we show.

4. Section 2.1 and 2.2 Spin-up of the land model is not mentioned. Is no spin-up required when land carbon and biogeographical changes are switched off?

Indeed, since the dynamics of land carbon and biogeography are not active in our simulations, the longest remaining time scales reside in soil memory. Because the AMIP simulations underlying our assessment start from a historical simulation of the related full Earth System Model, the soil water reservoirs are upon start already filled to a realistic level. We address this topic in our new section "Simulation setup" that was included in reaction to a comment by reviewer #1 (see line 131-139):

*"In case of ICON-A+JSBACHv4, data from year 1980 were taken for initialization from the respective historical simulation (Jungclaus et al., 2022), while the initialization of ECHAM6+JSBACHv3 is based on the state of the year 1979 of the associated historical CMIP6 simulation (Max Planck Institute for Meteorology, 2020). After model start the atmosphere equilibrates within days. Because land carbon and biogeographical components are not active in our AMIP simulations, the slowest land variable in this setup is soil moisture. By the initialization from a historical simulation, the soil water reservoirs are upon start already filled to a realistic*

*level. Soil water memory is typically a few months, only in desert regions it lasts up to a year (Hagemann and Stacke, 2015). But soil water memory stems (in our model) mostly from below the root zone (Hagemann and Stacke, 2015) so that it is largely decoupled from the active water cycle at the monthly scale that we consider for our assessment. "*

\* S. Hagemann and T. Stacke: Impact of the soil hydrology scheme on simulated soil moisture memory, Climate Dynamics, 44, 1731–750, 2015.

Minor comments:

- Please number equations.

Done.

- Around L205: GRACE data are used for evaluation of Terrestrial Water Storage (TWS). However, JSBACH does not include aquifers and their changes may influence changes in TWS from GRACE. The authors normalize the model and GRACE data and compare normalized, climatological month-to-month changes to account for these shortcomings. The authors should discuss the implicit assumption behind their approach and potential shortcomings. E.g., relative changes in aquifer water storage are assumed to have the same magnitude and phasing as TWS. How plausible is this?

We agree to the reviewer that we should be a bit more specific concerning the underlying assumptions. In two additions to the article's text we now point out that only the seasonal cycle of TWS is investigated and not long-term trends.

We added the following paragraph in line 217:

*"Here we are mainly interested in the question how JSBACHv4 performs in the context of climate and Earth System modelling. For the fluxes to the atmosphere a correct reproduction of the seasonal cycle in TWS is essential. Thus, only changes of TWS in the course of the year are analysed and not long-term trends. Our comparison is based on the assumption that the additional signal from the hydrology below 10 m that is present in the observational data but not in JSBACH does only negligible contribute to monthly TWS changes. We assume that below 10 m depth the hydrological processes are already so slow that they don't add to the phasing of the overall seasonal signal. This pertains in particular to potential signals from aquifers, whose recharging times are much larger than the monthly time scale considered in our comparison (typically decades to millennia)."*

We added the following paragraph in line 239:

*"Only the average seasonal changes in TWS are evaluated by this method. The amplitude of TWS variations as well as long-term trends in TWS are not considered. Therefore, a low value of $Q_{TWS}$*

*does not necessarily mean that variations in TWS are simulated realistically in a quantitative sense, but it shows that the seasonal phasing of TWS is captured by the model."*

- L249: I am a bit puzzled that NPP depends on fire. A more conventional definition is that NPP minus any carbon fluxes to the atmosphere from perturbations such as fire, herbivore grazing, pests, and mortality defines Net Biome Production (NBP). Then, carbon release by fire is not part of NPP as suggested here.

Indeed, without further comments our listing of fire as one of controls of NPP can only be misleading: In contrast to e.g. LAI or temperature, the effect of fire is only indirect and thus we better should not mention fire in the revised manuscript. We deleted fire in line 281.

But just to explain: wildfires indeed play a role for NPP because (as implemented in JSBACH) they reduce leaf carbon, thereby LAI, and thus lead to a reduction of productivity.

- Caption Fig. 1: typo Arctic, Antarctic

Done.

- Fig 7: typo: issignificant

Done.

- L380: typo: (5)-> (Fig. 5)

Done (now line 426).

---

## Author Response (AR2)

**Response to Report #2**

We thank both reviewers for revising our manuscript. In the following we address each of the comments of referee #3 point-by-point. We paste the original comments in red and our response in black. Our corresponding changes to the manuscript are set in italic.

**Report #2**

I was not one of the original reviewers of the MS but I have reviewed their comments, the authors' replies and the amended MS.

I think the MS would benefit from some English copy editing, which should be provided by the journal. It is a hard slog to get through the paper. Part of it is due to writing style.
We thank the reviewer for taking the burden! As non-native English speakers we cannot judge how much of the 'slog' is caused by unlucky formulations but would definitely appreciate suggestions for improvement, maybe this is a task for the journal's production team during the last stage of the publication process.

Generally the paper could be shortened as it contains a fair amount of discussion in the results section and thereby some overlap/repetition in the actual discussion section. Ideally a Results and Discussion section would be created along with a separate Conclusions. Although I would understand if appetite was low for such a large rewrite.
Assessment papers, like ours, are generally not very sexy and reading is typically over large parts quite boring (which may be another reason that working through it is a real 'slog'). Nevertheless, such papers are a necessity. To make reading a bit more interesting we partly added to the results section also some immediate discussion. As a consequence there is some overlap between the "Results" and "Discussion and Conclusions" sections. But this is by purpose: The whole assessment is meant as a reference for all scientists interested in and/or working with this model. Therefore we designed the "Discussion and Conclusions" section as a quick reference to the main findings. Thanks to the reviewer's comment we now realize that this purpose is not clear from the section title so that we changed it to *"Summary and Conclusions"*.

As you can see from my comments below I am quite critical of the chosen variables to assess the models with, as well as the use of single reference (observation-based) datasets to compare the models with. I would usually request that more datasets get incorporated as I view the use of only single datasets, when others exist readily, as not useful to truly evaluate a model. However in this MS I see the model results are sufficiently poor as to not make much difference. I don't mean that in a disparaging sense, most coupled models perform relatively poorly against reference datasets. Yet, I suggest for future papers to not rely upon single reference datasets unless only one really exists.
Our paper is indeed not a full evaluation of model performance. Instead it is meant as a first assessment documenting the plausibility of simulation results. As a global Earth system model, ICON-ESM is not meant to reproduce observations in detail, but only general patterns. Obvious reasons for this are the usage of global parametrizations and missing representation of much process detail. But a particular challenge for globally coupled models is the mutual imprint of model bias between separately developed large model components (atmosphere, land, ocean, cryosphere) that makes it necessary to tune components against bias in other components. Note that ICON-Land is not meant as a separate Dynamic Global Vegetation Model (DGVM) but developed to be the land component of an Earth system model. Thereby in the interest of an overall acceptable performance individual components must generally perform poorly against observations. For such models it is

therefore in particular in early stages of model development (ICON is brand-new!) part of the modelling strategy to aim only at reproducing system behaviour in the large. Assuming that different observational data sets for the same variable all reproduce the large scale patterns, the inclusion of further data sets would only in exceptional cases reveal further relevant insight into system behaviour and would in addition further reduce the readability of our paper.

You needn't cite this or anything, but I would point out that land surface models typically do poorly for LAI, e.g. see Seiler et al. 2022, so the model poor LAI is by no means unique.

*Seiler, C., Melton, J. R., Arora, V. K., Sitch, S., Friedlingstein, P., Anthoni, P., Goll, D., Jain, A. K., Joetzjer, E., Lienert, S., Lombardozzi, D., Luyssaert, S., Nabel, J. E. M. S., Tian, H., Vuichard, N., Walker, A. P., Yuan, W., and Zaehle, S.: Are terrestrial biosphere models fit for simulating the global land carbon sink?, J. Adv. Model. Earth Syst., 14, https://doi.org/10.1029/2021ms002946, 2022.*

We agree and think it makes sense to mention this as it gives context to the assessment. We included the sentence

*"This is not unique for our model (see Seiler et al., 2022)"*

in line 629 and the corresponding reference in line 863.

line 20 - what does 'weaker' FAPAR mean? Lower?

We added *"lower"* in line 21.

Why compare against NPP rather than the more directly observable GPP? GPP has at least four independent global products that could be compared to. NPP 'observations' on a global scale are always going to be a fairly derived/modelled product. Similar question for WUE, why not consider the ET and GPP themselves rather than a variable derived from them?

From the perspective of global carbon cycle modelling (which is next step in the development of ICON-ESM) one must get NPP right, because its NPP not GPP that determines carbon storage. Indeed, a precondition to get NPP right is to obtain proper values for GPP and autotrophic respiration. And since GPP data products are from an observational perspective more reliable than NPP products we understand the reviewer's suggestion. Nevertheless, for the purpose of our paper to document the present model development stage also in view of further model development towards inclusion of the global carbon cycle we can at the moment easily do without GPP but not without NPP. And the paper is already quit long so that we decided to not consider GPP in addition. Concerning WUE, the situation is quite similar. From a modelling perspective it is for a first assessment more interesting to get insight into an integrated quantity like WUE than its individual components.

Lines 53 - 55 seems to imply that the land carbon cycle is switched off(?), if so, how do you do NPP? Even after fully reading the paper I still found this part puzzling. The paper seems to suggest no C cycle then it proceeds to compare LAI (so leaves made from C) and NPP (C again).

We understand that the reviewer is puzzled: The JSBACH phenology model is not carbon based but rather phenomenological describing the development of the LAI by a combination of phases of logistic growth towards a prescribed PFT-specific maximum LAI and exponential drop of leaves. And the onset and end dates of theses phases are triggered by the local environmental conditions (temperature, relative soil moisture). Because of this peculiarity of the JSBACH phenology model the growth and maintenance respiration cannot be determined from the amount of allocated carbon in living tissues. Instead the calculation of these respiration terms follows the implementation of the BETHY model [1], where they are calculated as a prescribed fraction of GPP, in exceptional cases modified by prescribed limits to allocation (Reick et al., 2021). This is the reason why NPP belongs as the other variables assessed in our paper to the 'fast' processes of JSBACH that can be assessed without considering carbon storage in tissue pools. To prevent this confusion, we replaced the imprecise wording "land carbon cycle" in line 55 so that the sentence now reads

"*These variables represent processes that are fast compared to the cycling of carbon between land storage pools or climate induced biogeographical changes in landcover.*".
Moreover, we added to the NPP section 2.3.6 of the manuscript the sentence:
"*Note that autotrophic respiration is calculated as a prescribed fraction of GPP instead from carbon allocated in the different plant tissues. Accordingly, NPP belongs as the other variables considered here to the 'fast' variables of JSBACH that can be assessed without consideration of the cycling of carbon between different storage pools.*"
in line 295.

[1] W. Knorr, Annual and Internannual $CO_2$ Exchanges of the Terrestrial Biosphere: Process-Based Simulations and Uncertainties, Global Ecology and Biogeography 9 (2000), 225-252.

Line 60 - this point about making sure the implementation is free of defects could actually go in the abstract. Otherwise the last line of the abstract sounds a bit optimistic since the rest of the abstract seems to suggest that v3 and v4 share many of the same biases, which makes a reader wonder why there is so much optimism in the abstract's last sentence. To be more plain - reading the abstract without knowing that v3 and v4 are almost identical except for the framework implemented in makes one think the problems in JSBACH must be really stubborn to not improve much between versions! This point seems to have confused Ref #2 as their first comment implies they missed that one of the main motivations for this paper is to ensure the implementation was correct.
True. As suggested we include this point in the abstract in line 23 by replacing its last sentences by the new formulation:
"*Overall, the biases found in the different assessment variables are either already known from the previous implementation in MPI-ESM1.2, or have changed because of the coupling with the new atmospheric component ICON-A. As discussed, there is a good perspective to mitigate these biases by an improved processes representation. Accordingly, this study demonstrates the technically successful completion of the re-implementation of JSBACH into ICON-ESM-V1.*"

Table 1 should list number of snow layers so the older versions two layers can be listed alongside the newer versions five.
True. We listed the number of snow layers now in table 1.

line 142 - how can wood turnover not be implemented but you can calculate NPP? Doesn't the NPP contribute to plant tissues which would need to turnover to present a continual buildup. I still don't get this part.
Please see our answer to your comment on lines 53-55 of our manuscript.

line 198 - 'skin reservoir on the surface' = ponded water?
 "Skin reservoir" is the hydrological terminus technicus for the storage pool of water from precipitation and dew on the canopy and the ground (see e.g. [2]). However, to avoid confusion about this term we changed the sentence to
"*In JSBACH, TWS is the sum of water stored as snow and water on the surface and the canopy, as soil water and soil ice, and as runoff.*"
in line 200.

[2] P. Viterbo, L. Illari (1994). The impact of changes in the runoff formulation of a general circulation model on surface and near-surface parameters, Journal of hydrology, *155*(3-4), 325-336.

line 199 - 'as or in' - fix, confusing as is.
We deleted "or in".

line 244 - 'a high LAI typically reduces albedo', really? Grass with an LAI of 2 will still be lighter (higher albdeo) than needle leaf evergreen forest with LAI of 4 so I am not sure what is meant there.
Sorry, but as far as we see your example confirms what we are saying: a high LAI (needle leaf evergreen forest with 4) has a lower albedo than a low LAI (grass with 2). Nevertheless, we agree that this statement is too bold: how albedo is affected depends not on the LAI alone but also on the albedo of the underlying ground and the presence of snow ('snow masking'). Therefore increasing LAI may, depending on the situation, raise or lower the albedo. We thus drop the reference to albedo in the respective sentence and keep only the remarks on transpiration and productivity. The modified sentence thus reads:
*"A high LAI typically enhances transpiration of water and in general also primary productivity."*
in line 247.

Why compare against only one global LAI product? Is there reason to suppose that product is without biases? Similar question about the other variables. It is not sufficient to compare against only one observation-based dataset if other reliable ones are available. To do otherwise assumes that all 'observations' are without bias - a naive assumption at best.
We agree that it would indeed be naive to assume that "all 'observations' are without bias". But as already mentioned above, our modelling strategy is not to get simulation results correct in detail, but only in the large. Our assumption is that for this purpose any generally accepted observational product will do.

Why move from two snow layers to five? I realize the full logic might be outside the scope of this paper but it puzzles me as to what would be gained. Is it just to have finer discretization?
Yes, we intended a better representation of the heat fluxes into the ground. Anyhow, the term two layer for snow on/in the soil and on the canopy is misleading. Actually there is only one snow layer on the soil and the snow on the canopy is addressed separately in our paper. Therefore we changed "two" to *"one"* in line 132 and in table 1.

Figure 1 is a nice inclusion
Thanks.

Fig 6 - does 'year' in the top two plots stand for annual mean? Perhaps make that more clear. Same in fig 7, table 4, and perhaps others.
We replaced "year" in all tables and figures with *"annual"*.

line 409 - RSM is not defined prior to use.
We realized that the "relative" it not necessary in the term precipitation-RSM feedback. Therefore we replaced RSM in line 418 with *"soil moisture (SM)"* and "precipitation-RSM feedback" with *"precipitation-SM feedback"* throughout the paper. We included *"Relative Soil Moisture"* at its first appearance in line 469.

line 414 - it is easy to miss that the 'general rule' comment is for January, I would suggest rewording to make clear it is not intended to be a year round phenomenon.
We changed this sentence to
*"However, for both models a higher TCC has a warming effect in January – as expected from the general rule that a cloud cover tends to warm the surface in winter (Chen et al., 2000)."*
in line 422.

Fig 11 - max amount of water possible above wilting point? Is this field capacity or ? It is better to use more standard terms as the max amount of water possible above wilting could include inches of water ponded on the surface - the definition given is not excluding that.

Yes, 'maximum amount of soil water' is field capacity. We thus reformulated the respective sentence in the caption of Fig. 11 into

*"RSM is calculated as the ratio between the amount of soil water above wilting point and the amount of water between field capacity and wilting point. RSM thus characterizes the soil conditions concerning water stress of plants (the plant usable field capacity)."*

line 513 - 'The above named opposing northern mid latitudes NPP biases'. Consider revising for clarity, since it is new para it is hard to know what the above named is.

We reformulated the respective sentence into:
*"The shift in NPP bias from JSBACHv3 to JSBACHv4 seen in northern mid latitudes in July has several reasons."*
now in line 521.

L 562 - fix ref to appendix
Done.

---

## Author Response (AR3)

**Response to the Topical Editor**

Could you provide the permanent link for the code and data such as zenodo, not doi only?

We added direct links to the doi in line 678 and 684. To put the data also on zenodo will not be possible as this would be a bigger decision for the institute as a whole.